# Greedy Information Projection for LLM Data Selection

## Abstract

We present *Greedy Information Projection* (GIP), a principled framework for choosing training examples for large language model fine-tuning. GIP casts selection as maximizing mutual information between a compact subset of examples and task-specific query signals, which may originate from LLM quality judgments, metadata, or other sources. We formulate a mutual information framework from data and query embeddings, the objective has a closed form and naturally balances quality and diversity. We show that optimizing this objective is equivalent to maximizing the projection of the query embedding matrix onto the span of the selected data, yielding a geometric explanation for the co-emergence of quality and diversity. Building on this view, we develop a fast greedy matching-pursuit procedure with efficient projection-based updates. On instruction-following and mathematical reasoning datasets, GIP selects compact subsets that match full-data fine-tuning while using only a small fraction of examples and compute, unifying quality-aware and diversity-aware selection for efficient fine-tuning.

## 1 Introduction

A central question in the discussion of large language model (LLM) training is how one should select data. With recent developments of large-scale pretrained language models such as GPT-4 (Achiam et al., 2023), Qwen (Qwen et al., 2025) and Llama (Grattafiori et al., 2024), the community has witnessed models achieving remarkable performance across a broad spectrum of natural-language tasks. Typically, these models train on large amounts of data that scale with the model size, commonly known as the scaling law. Empirical scaling laws of LLM reveal that model capability grows predictably with (i) parameter count, (ii) compute, and (iii) training data volume (Hoffmann et al., 2022; Kaplan et al., 2020). Yet practitioners increasingly observe a second, subtler regime: once data quantity is sufficient, *data quality* becomes the primary factor limiting further gains. Consequently, a fundamental challenge in both pre-training and instruction tuning is to select a subset of samples that maximizes downstream performance while respecting resource constraints.

Earlier work addressed the problem through diverse dataset construction (Wang et al., 2022; Taori et al., 2023b) and ad-hoc filtering heuristics such as perplexity thresholds, deduplication, or clustering (Bukharin et al., 2024; Zhao et al., 2024; Chen et al., 2023b; Ge et al., 2024). While effective, these heuristics offer little theoretical guidance on *why* a particular example is valuable, and they do not unify quality and diversity under a single objective.

**This work.** We present a principled *information-theoretic* framework for data selection built on a mutual information (MI) metric between Gaussians parameterized by data and query embeddings. As we will demonstrate later, this metric promotes both diversity and quality in a single objective, and also gives rise to efficient algorithm design. Our main contributions are:

1. **Principled theoretical formulation.** We propose a principled theoretical framework that casts the data selection problem as maximization of mutual information. Optimizing this objective naturally promotes diversity and quality of selected data. This framework is also flexible, enabling balanced selection with respect to disparate information sources, such as factuality and style scores from LLM evaluations.

2. **Efficient approximation algorithms.** We develop a greedy matching pursuit (MP) approximation algorithm that solves an approximate dual problem. The Greedy MP approach

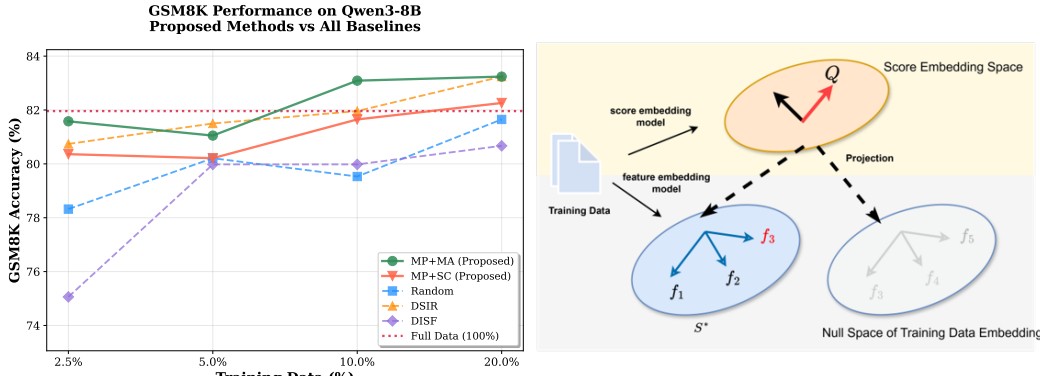

Figure 1: **Left:** GSM8K performance comparison on Qwen3-8B across different training data percentages (2.5%, 5%, 10%, 20%). Our proposed methods MP+MA and MP+SC consistently outperform baselines (Random, DSIR, DISF) and approach or exceed full dataset (100%) performance with only 10-20% of training data, demonstrating significant data efficiency. **Right:** Geometric interpretation of GIP. The method maximizes mutual information between Gaussians parameterized by data embedding matrix $F$ and score embedding matrix $Q$. This is equivalent to minimizing the volume (determinant) of score embeddings projected onto the null space of selected data, naturally balancing quality (high-score items) and diversity (new directions in embedding space).

       scales linearly with the total size of available data in practice, enabling data selection under realistic budget constraints.

3. **Strong empirical results.** On instruction-tuning benchmarks, our *Greedy Information Projection* (GIP) achieves substantial data efficiency gains over state-of-the-art baselines. GIP matches or exceeds full dataset performance using only 1-20% of training data across MT-Bench, BBH, and GSM8K benchmarks—demonstrating up to $5\times$ data reduction; see Figure 1.

## 2 RELATED WORK

### 2.1 DATA CURATION FOR LARGE-SCALE LANGUAGE MODELS

Diversity and quality are recognized as crucial factors impacting the overall quality of model training (Sener & Savarese, 2018; Chen et al., 2024; 2023a). With the recent development of large language models, there is a growing trend in studies focusing on data composition quality and diversity, and how these factors influence language model pretraining and instruction tuning.

For example, Lee et al. (2022) highlight the importance of diversity in improving training efficiency and propose a method to remove near-duplicate records from training data. Similarly, Bukharin et al. (2024) emphasize the significance of both quality and diversity in instruction tuning data, proposing a general method to balance these two aspects in training data selection. Furthermore, Du et al. (2023) introduce a systematic data selection approach that combines data quality, diversity, and augmentation for instruction tuning. However, none of Bukharin et al. (2024); Du et al. (2023) have a unified framework to unify *both* quality and diversity; diversity was either applied as a penalty or considered in a different stage of data selection.

Recent work has focused on more sophisticated data selection strategies. Chen et al. (2023b) propose instruction-following difficulty (IFD) scoring to identify high-quality instruction data. Ge et al. (2024) introduce clustering-based active retrieval (CaR) that uses representativeness and uncertainty for data selection. Xia et al. (2024) develop LESS, a gradient-based method for selecting informative training data. Xie et al. (2023) propose data selection using importance resampling (DSIR) for domain adaptation. Our work differs by providing a unified information-theoretic framework that

naturally balances quality and diversity in a single objective, while being computationally efficient and theoretically grounded.

## 2.2 Information-Theoretic Objectives in Selection and Clustering

There is a long history of applications of mutual information to data selection and clustering. Mutual information can capture complex relationships between features that might not be apparent (Knops et al., 2006). Mutual information is also flexible, working on various types of data such as categorical data (He et al., 2008) and numerical data (Kraskov et al., 2005). Approximations of mutual information has also been applied to k-means (Sugiyama et al., 2014; Calandriello et al., 2014). More recently, mutual information based algorithms have been developed for clustering (Do et al., 2021), community detection (Newman et al., 2020), 3d object representation learning (Sanghi, 2020), and unsupervised sentence embeddings (Zhang et al., 2020). Theoretical developments of mutual information inequalities have also been applied in pairwise comparisons (Lee & Courtade, 2021) and generalized linear models (Lee & Courtade, 2020), where optimal selections and minimax algorithms are understood to be closely related to singular values of query matrices; see, e.g., (Lee, 2022) for a broad discussion.

It is important to note that mutual information is often difficult to compute and various forms of estimators and approximations are used to reduce the computation cost; see (Kraskov et al., 2004) for a classic mutual information estimator. In this paper, we consider the mutual information of joint-Gaussian variables — Gaussianity provides us many favorable elementary expressions which allow us to bypass much of the complexity of estimating mutual information.

## 2.3 Active Learning and Coreset Selection

Our work is related to active learning (Settles, 2009) and coreset construction (Bachem et al., 2017), which aim to select informative examples and compact subsets for downstream tasks, respectively (e.g., DSIR (Xie et al., 2023) and DISF(Fan et al., 2025)). We build on this broader theme of data efficiency, introducing an information-theoretic selection criterion that jointly considers quality and diversity in the instruction-tuning setting.

## 3 Problem formulation

We are given a set of $m$ data points $\mathcal{F} = \{f_1, \ldots, f_m\}$ with $f_i \in \mathbb{R}^d$ representing features of selectable data, and $n$ score vectors $\mathcal{G} = \{g_j \in \mathbb{R}^m, j \in [n]\}$ where each score vector $g_j$ represents evaluations on all $m$ data points for a specific quality dimension. These scores can come from LLM evaluators assessing quality dimensions (helpfulness, accuracy, reasoning) or internal measures like self-consistency estimates.

Our goal: Given data features $\mathcal{F}$ and quality scores $\mathcal{G}$, select at most $k$ data points that are both *diverse* (spanning different regions of the feature space) and *high-quality* (aligned with the provided scores).

## 3.1 Mutual Information Formulation

Let $F \in \mathbb{R}^{d \times m}$ be the data embedding matrix with $F = [f_1, f_2, \ldots, f_m]$, and let $G \in \mathbb{R}^{m \times n}$ be the score matrix with $G = [g_1, g_2, \ldots, g_n]$. For selection $S \subseteq [m]$ with $|S| \leq k$, define $F_S \in \mathbb{R}^{d \times |S|}$ as the matrix of selected data columns. We assume data embeddings are normalized: $\|f_i\|_2 = 1$ for all $i \in [m]$.

Our framework requires query embeddings $Q \in \mathbb{R}^{d \times n}$ such that $F^\top Q \approx G$. This establishes a connection between the feature space (where diversity is measured) and the score space (where quality is defined).

**Regularization and stability.** Since $m \gg d$ in practice, $F$ typically has rank at most $d$ and is not full row rank. To ensure existence and numerical stability of $Q$, we use Tikhonov regularization. Specifically, we solve:

$$Q_\epsilon = \arg\min_Q \|F^\top Q - G\|_F^2 + \epsilon \|Q\|_F^2 \tag{3.1}$$

with solution $Q_\epsilon = F(F^\top F + \epsilon I_m)^{-1}G$ for small $\epsilon > 0$. This ensures $\|F^\top Q_\epsilon - G\|_F \leq C\epsilon$ for some constant $C$. Details are provided in Appendix D.

Consider standard Gaussian $Z \in \mathcal{N}(0, I_d)$ and transformations $Z_Q := Q^\top Z$, $Z_{F_S} = F_S^\top Z$: Remark (Gaussianity as a modeling device). We do not assume that raw data or query embeddings are Gaussian. Gaussianity enters only via the auxiliary variable $Z \sim \mathcal{N}(0, I_d)$, used to derive a closed-form, rotation-invariant mutual information surrogate based on the linear images through $F_S$ and $Q$.

$$\begin{bmatrix} Z_Q \\ Z_{F_S} \end{bmatrix} \sim \mathcal{N}\left(0, \begin{bmatrix} Q^\top Q & Q^\top F_S \\ F_S^\top Q & F_S^\top F_S \end{bmatrix}\right) \tag{3.2}$$

The mutual information between $Z_Q$ and $Z_{F_S}$ is:

$$I(Z_Q; Z_{F_S}) = \frac{1}{2} \log\left(\frac{\det(Q^\top Q)\det(F_S^\top F_S)}{\det(\Sigma)}\right) \tag{3.3}$$

where $\Sigma$ is the joint covariance matrix in equation 3.2.

**Theorem 1.** *The mutual information maximization is equivalent to:*

$$\arg\max_S I(Z_Q; Z_{F_S}) = \arg\min_S \det\left(Q^\top \left(I - F_S(F_S^\top F_S)^{-1}F_S^\top\right)Q\right) \tag{3.4}$$

**Intuition.** The matrix $P_S := I - F_S(F_S^\top F_S)^{-1}F_S^\top$ projects $Q$ onto the null space of $F_S$. Minimizing $\det(Q^\top P_S Q)$ selects $S$ that minimizes the volume of $Q$ after projection, naturally encouraging both diversity (expanding into new directions in embedding space) and quality (aligning with high-score directions).

**Theorem 2** (Quality Bounds). *For the MI-optimal selection $S^*$ from Theorem 1, there exists $\delta_{S^*}$ such that*

$$\|F_{S^*}^\top Q\|_2 \geq \delta_{S^*}\sqrt{1 - \left(\frac{\eta}{\det(Q^\top Q)}\right)^{1/r}} \tag{3.5}$$

*where $\eta = \det\left(Q^\top(I - P_{F_{S^*}})Q\right)$ and $r$ is the rank of $Q$.*

**Quality Guarantee.** This theorem provides a lower bound on how well the selected data $F_{S^*}$ aligns with the query directions $Q$. The bound shows that our MI-optimal selection maintains quality.

When scores are missing ($Q = \emptyset$), we maximize the entropy $h(Z_{F_S}) = \frac{1}{2} \log \det(F_S^\top F_S)$, which encourages pure diversity consistent with our framework.

## 4 GREEDY APPROXIMATION ALGORITHM

With small data sizes, one can permute through all possible selections to obtain the optimal selection $S$, but the exponential computation cost is impractical in real-world scenarios where data sizes are large. Instead, we propose to optimize equation 3.4 by approximations.

We employ an approximation by optimizing an upper bound of equation 3.4 that follows by a direct application of the classic AM-GM inequality.

**Theorem 3.** *Given $Q = [q_1 \quad \dots \quad q_n]$, the determinant stated in equation 3.4 satisfies*

$$\det\left(Q^\top \left(I - F_S(F_S^\top F_S)^{-1}F_S^\top\right)Q\right) \leq \left(\frac{\mathrm{Tr}((Q^\top \left(I - F_S(F_S^\top F_S)^{-1}F_S^\top\right)Q)}{n}\right)^n. \tag{4.1}$$

By shifting our optimization target to the right-hand side of equation 4.1, we loosen the optimization objective to the trace, which enjoys linearity. In particular, recall that $Q$ is a $\mathbb{R}^{d \times n}$ matrix constructed by a concatenation of $q_1, \dots, q_n \in \mathbb{R}^d$. By linearity, we have

$$\mathrm{Tr}((Q^\top \left(I - F_S(F_S^\top F_S)^{-1}F_S^\top\right)Q) = \sum_{i=1}^n \mathrm{Tr}\left(q_i^\top \left(I - F_S(F_S^\top F_S)^{-1}F_S^\top\right)q_i\right), \tag{4.2}$$

and we obtain a quadratic expression with favorable properties to work with. Note also that the minimization of equation 4.2 over $S$ depends on $Q$ only through scores $g_i := F^\top q_i \in \mathbb{R}^n$, $i \in [n]$. Therefore, the expression in equation 4.2 permits us to work directly with scores $G$ *without the knowledge of $Q$*.

In this section, we introduce our **Greedy MP** algorithm that approximates the *dual problem* by a matching pursuit approach.

### 4.1    GREEDY MATCHING PURSUIT (MP)

The intuition is to note that equation 4.2 involves computing lengths of $q_i$ projected onto the null space of $F_S$, which can be written as a dual form:

$$\min_S \sum_{i=1}^n \min_{\lambda \in \mathbb{R}^k} \left\| q_i - \sum_{j \in S} \lambda_j f_j \right\|_2^2. \tag{4.3}$$

Greedy MP solves this by minimizing residual gain across all query embeddings: at step $t + 1$,

$$s_{t+1}, \lambda_{t+1}^* = \arg \min_{s \in [m]/S_t} \sum_{i=1}^n \min_{\lambda \in \mathbb{R}^k} \| r_i - \lambda_i f_s \|_2^2, \tag{4.4}$$

where $r_i = q_i - \sum_{j=1}^t \lambda_j^* f_{s_j}$ is the residual of the $i$-th score at step $t+1$. Note that for each individual $q_i$ and candidate $f_s$, the minimizer $\lambda_i^* = r_i^\top f_s$ is unique, and satisfies

$$\min_{\lambda \in \mathbb{R}} \| r_i - \lambda f_s \|_2^2 = \left\| r_i - r_i^\top f_s f_s \right\|_2^2 = \| r_i \|^2 - (r_i^\top f_s)^2. \tag{4.5}$$

By substituting equation 4.5 into equation 4.4, we see that the greedy selection of $s_{t+1}$ is simply one that solves

$$s_{t+1} = \arg \max_{s \in [m] \setminus S_t} \sum_{i=1}^n (r_i^\top f_s)^2.$$

By defining *residual score matrix* $W \in \mathbb{R}^{n \times m}$ where $W = [r_1 \ldots r_n]^\top [f_1 \ldots f_m] = \begin{bmatrix} r_1^\top f_1 & \cdots & r_1^\top f_m \\ \vdots & \ddots & \vdots \\ r_n^\top f_1 & \cdots & r_n^\top f_m \end{bmatrix}$, we get $s_{t+1}$ by solving

$$s_{t+1} = \arg \max_{s \in [m] \setminus S_t} \sum_{i=1}^n (W_{i,s}^{(t)})^2.$$

After selection $s_{t+1}$, $W$ can be updated with

$$W_{i,j}^{(t+1)} \leftarrow W_{i,j}^{(t)} - \Phi_{j,s_{t+1}} \cdot W_{i,s_{t+1}}^{(t)}, \tag{4.6}$$

where $\Phi_{j,s} = f_j^\top f_s$ are the precomputed inner products between data vectors. This update corresponds to the new residual $r_i' = r_i - (r_i^\top f_{s_{t+1}}) f_{s_{t+1}}$, yielding $r_i'^\top f_j = r_i^\top f_j - (f_j^\top f_{s_{t+1}}) r_i^\top f_{s_{t+1}}$ as expected.

Immediately, this suggests we can efficiently solve equation 4.3 by maintaining and updating a residual score matrix $W$ *in-place*, while inner products of data vectors can be efficiently looked up by precomputing $F^\top F$ in memory.

**Algorithm Initialization.** We initialize $W^{(0)} = G = Q^\top F \in \mathbb{R}^{n \times m}$, where $G_{ij} = q_i^\top f_j$ represents the initial correlation between the $i$-th query and $j$-th data point. At iteration $t = 0$, we have $r_i^{(0)} = q_i$ for all $i \in [n]$, so $W_{ij}^{(0)} = r_i^{(0)\top} f_j = q_i^\top f_j = G_{ij}$.

**Algorithm Variables.** In Algorithm 1: $F \in \mathbb{R}^{d \times m}$ is the data matrix with columns $f_j$; $G \in \mathbb{R}^{n \times m}$ is the score matrix where $G = Q^\top F$; $W^{(t)} \in \mathbb{R}^{n \times m}$ tracks residual correlations $r_i^{(t)\top} f_j$ at iteration $t$; $\Phi \in \mathbb{R}^{m \times m}$ stores precomputed data inner products $f_i^\top f_j$; and $S$ accumulates the selected indices.

---

**Algorithm 1** Greedy matching pursuit (MP)

---

**Require:** Data matrix $F \in \mathbb{R}^{d \times m}$, score matrix $G \in \mathbb{R}^{n \times m}$ (where $G = Q^\top F$), number of selections $k$
**Ensure:** Selection set $S$
1: Initialize $S = \emptyset$, $W^{(0)} \leftarrow G$ {$W \in \mathbb{R}^{n \times m}$}
2: Precompute $\Phi = F^\top F \in \mathbb{R}^{m \times m}$ {Data inner products}
3: **for** $t = 1$ to $k$ **do**
4:     $s_t \leftarrow \arg \max_{j \in [m] \setminus S} \sum_{i=1}^{n} (W_{i,j}^{(t-1)})^2$ {Select best candidate}
5:     $S \leftarrow S \cup \{s_t\}$
6:     **for** $i = 1$ to $n$, $j = 1$ to $m$ with $j \notin S$ **do**
7:         $W_{i,j}^{(t)} \leftarrow W_{i,j}^{(t-1)} - \Phi_{j,s_t} \cdot W_{i,s_t}^{(t-1)}$ {Update residuals}
8:     **end for**
9: **end for**
10: **return** Selection set $S$

---

**Analysis of relaxation** Although the greedy algorithm optimizes the relaxed objective, we studied its approximation with respect to the original objective. We found that on controlled instances where we can enumerate the optimum, the linearization tracks the original objective (4.2) well. (Appx. C, Tab. 9).

The algorithm is attached in Algorithm 1. Notably MP algorithms have been broadly studied and applied since the nominal work of Mallat & Zhang (1993). Here, we contribute a new variation of MP for data selection.

## 4.2 COMPUTATIONAL COMPLEXITY AND PRACTICAL COSTS

The Greedy MP algorithm has a total runtime complexity of $O(m^2 d + mnk)$, where $m$ is the total number of data points, $n$ is the number of scores, and $k$ is the desired subset size. This breaks down as follows: (1) Precomputation of $\Phi = F^\top F$ requires $O(m^2 d)$ operations, performed once; (2) Initialization of $W^{(0)} = G$ is $O(mn)$ if $G$ is provided, or $O(mnd)$ if computed from $Q^\top F$; (3) For each of the $k$ iterations, we select the best candidate ($O(mn)$) and update the residual matrix $W$ for $O(m)$ entries ($O(mn)$), giving $O(mnk)$ for the iterative process. The memory complexity is $O(m^2 + mn)$ for storing $\Phi$ and $W$. In practice, since $n$ is typically small, the algorithm scales nearly linearly with the dataset size $m$ after the initial precomputation, making it practical for large-scale data selection.

We summarize the selection-phase runtime and resource requirements of the proposed Greedy MP against common baselines used in instruction/data selection. The focus is on asymptotic behavior with respect to: $m$ (candidate pool size), $k$ (selected subset size), $d$ (embedding dimension), $m_{\text{val}}$ (validation set size), and $n$ (number of query/task score vectors; typically small). We contrast whether each method requires (a) a trained model for gradients/logits, and (b) an external validation subset.

| Method | Selection Time Complexity | Validation Data? | Model Training? |
|---|---|---|---|
| MP (Ours) | $O(mk)$ | No | No |
| LESS | $O(m \, m_{\text{val}} \, d)$ | Yes | Yes |
| DISF | $\tilde{O}(mkd^2)$ | Yes | Yes |
| DSIR | $O(md + m_{\text{val}}d)$ | Yes | Yes |

Table 1: Selection-phase complexity and requirements. $n$ is typically small ($\ll m$). $\tilde{O}$ hides polylog factors.

Overall, the proposed methods yield strictly lower operational friction (no gradients, no held-out scoring) while matching or exceeding downstream performance (see Experiments). For detailed robustness and resource measurements (RAM usage / wall-clock), refer to Appendix A.

## 5 EXPERIMENTS

### 5.1 DATASETS AND BASELINE MODELS

**Training dataset.** We use the Alpaca 52k dataset which contains 52,000 diverse instructions and demonstrations in English (Taori et al., 2023b). This data is commonly used as a benchmark training dataset; see, for example, (Zhao et al., 2024; Bukharin et al., 2024; Ge et al., 2024). We apply Greedy MP to select training subsets of varying sizes: 512 samples (1% of full data) and 1000 samples (2% of full data) for MT-Bench and BBH evaluation, and subsets of 2.5%, 5%, 10%, and 20% of the full dataset for GSM8K evaluation.

**Baseline methods.** We compare against several state-of-the-art data selection methods:

- **Random**: Randomly selected subsets (standard baseline)
- **Full**: Training on the complete dataset (upper bound reference)
- **LIMA** (Zhou et al., 2023): High-quality manually curated examples
- **Alpagasus** (Chen et al., 2023b): Instruction-following difficulty (IFD) based selection
- **CaR** (Ge et al., 2024): Clustering-based active retrieval using representativeness and uncertainty
- **DSIR** (Xie et al., 2023): Data selection using importance resampling (for domain adaptation)
- **DISF** (Fan et al., 2025): gradual and informative data selection that iteratively surfaces the most valuable training examples
- **LESS** (Xia et al., 2024): gradient similarity-based efficient data selection

For mathematical reasoning evaluation on GSM8K, we include all eight baseline methods. For MT-Bench and BBH, we focus on the five most relevant baselines. All baseline methods use identical experimental settings and are retrained from scratch to ensure fair comparison.

**Base models.** We use three pre-trained language models for our experiments: Mistral-7B (Jiang et al., 2023), Qwen3-8B (Yang et al., 2025), and Qwen-4B (Yang et al., 2025). All models undergo supervised fine-tuning with training details provided in the supplementary material.

**Evaluation datasets.** We evaluated trained models on three comprehensive benchmark datasets: MT-Bench dataset (Zheng et al., 2023a), BIG-Bench Hard (BBH) (Suzgun et al., 2022), GSM8K (Cobbe et al., 2021). For MT-Bench, we use the standard MT-Bench evaluation protocol (Zheng et al., 2023a) where models generate responses based on VLLM (Kwon et al., 2023) to multi-turn conversations, and responses are scored by GPT-4.1 [1] with score range from 0 to 10. For BBH and GSM8K, we reported the accuracies based on the given ground-truth answer.

### 5.2 IMPLEMENTATION

We run experiments with the Greedy MP algorithm (Algorithm 1) as follows: **Data embeddings** $F$. The Alpaca 52k dataset is a generic instruction-response dataset that includes training data with instruction, input (context), and answer (response) triplets. For each triplet, we concatenate the components into a single sentence [*Instruction; Context; Response*] as the input sentence. We then use pretrained ModernBERT-Base (Warner et al., 2024) as the embedding model and use outputs from the final hidden representation layer as embeddings. The context length is set to 2048, aligned with the training context window length. We applied similar setting to GSM8K dataset, where we consider the question as instruction with context.

We consider two methods of generating score vectors: self-compression and LLM evaluation.

**Scores via self-compression.** Given embeddings $F \in \mathbb{R}^{d \times m}$ where each column $f_i$ represents the embedding of sample $i$, we compute a quality score for each sample based on its alignment with the entire dataset. We calculate the dataset alignment score:

$$g_i = \sum_{j=1}^{m} f_i^\top f_j = (F^\top F \mathbf{1}_m)_i,$$

---

[1] https://openai.com/index/gpt-4-1/

Table 2: Performance on MT-Bench and BBH benchmarks. Best results in **bold**. Detailed results with standard errors are provided in Appendix Table 16.

| Method | Data | Mistral-7B | | Qwen3-8B | | Qwen-4B | |
| --- | --- | --- | --- | --- | --- | --- | --- |
| | | MT-Bench | BBH | MT-Bench | BBH | MT-Bench | BBH |
| Full | 100% (52K) | 3.89 | **58.4** | 4.62 | 76.7 | 4.11 | 73.2 |
| Random | 2% (1K) | 3.84 | 57.4 | 6.25 | 72.3 | 5.14 | 69.4 |
| LIMA | 2% (1K) | 3.55 | 55.6 | 5.88 | 75.8 | **5.23** | 73.3 |
| Alpagasus | 2% (1K) | 3.83 | 56.5 | 6.14 | 71.9 | 4.93 | 63.6 |
| CaR | 2% (1K) | 3.59 | 57.4 | 5.78 | 76.2 | 3.92 | 73.8 |
| MP+MA | 2% (1K) | 3.92 | 57.6 | 6.25 | **80.5** | 4.91 | 74.3 |
| MP+MA | 1% (512) | **4.28** | 56.6 | **6.68** | 79.1 | 5.19 | 68.0 |
| MP+SC | 2% (1K) | 3.77 | 57.7 | 5.50 | 79.1 | 4.55 | **74.4** |
| MP+SC | 1% (512) | 4.14 | 57.4 | 6.26 | 76.5 | 4.93 | 73.9 |

where $\mathbf{1}_m \in \mathbb{R}^m$ is the all-ones vector. This score measures how well sample $i$ aligns with the overall dataset structure - samples with higher scores are more representative of the data distribution and thus more informative for training. The formulation is equivalent to computing the $i$-th row sum of the Gram matrix $F^\top F$, providing a centrality measure in the embedding space. This provides an *internal* information source where all signals come from the training dataset itself, without external supervision. Experiments using this self-compression scoring are labeled as MP+SC.

**Scores via LLM assessments.** We use GPT-4o [2] to generate evaluations between 0 to 5 in four criteria: coherence, helpfulness, accuracy, and difficulty, and use the total score across all four criteria as the score vector. The prompt is included in the supplementary material. Experiments using LLM quality assessments to obtain multi-attribute information are labeled as MP+MA.

**Baseline implementation details.** To ensure fair comparison, all baseline methods are implemented using identical experimental settings (see Table Appendix G.3 for details). For consistency, we use the same data preprocessing, embedding models, and evaluation protocols across all methods:

- **DSIR**: We implement the importance resampling approach from (Xie et al., 2023) using the target domain distribution as the reference for reweighting sample importance. Typically as the method needs a target domain dataset to compute the importance weights, we sample from test data as the target domain. Therefore DSIR could be considered as a strong baseline with oracle for GSM8K task.
- **DISF**: We implement the diversified file selection algorithm (DISF) from (Fan et al., 2025), starting from a random set and iteratively adding samples that maximize the diversity among single batch.
- **LESS**: We implement the LESS method from (Xia et al., 2024), we start with building gradient features from the base model. Similar as DSIR, we used the same validation set to compute the influence score for the training data, we then select the most influential data as in (Xia et al., 2024)

All baselines use identical LoRA fine-tuning configurations, learning rates and training epochs specified in Table Appendix G.3. This ensures that performance differences reflect data selection quality rather than training procedure variations.

## 5.3 MAIN RESULTS

Table 2 presents a comprehensive comparison between Greedy MP methods and baseline approaches across three model architectures on MT-Bench and BBH benchmarks. The results demonstrate the effectiveness of our proposed methods across different model scales and evaluation scenarios.

**MT-Bench Performance:** Our MP+MA method consistently achieves strong performance across all models. Notably, with 1% training data (512 samples), MP+MA achieves 4.28 on MT-Bench for

---

[2]https://openai.com/index/hello-gpt-4o/

Table 3: GSM8K performance across models and data percentages.

| Method | Qwen3-8B | | | | Mistral-7B | | | | Qwen-32B | | | |
|---|---|---|---|---|---|---|---|---|---|---|---|---|
| | 2.5% | 5% | 10% | 20% | 2.5% | 5% | 10% | 20% | 2.5% | 5% | 10% | 20% |
| Full | 81.96 | | | | 52.46 | | | | 87.64 | | | |
| Random | 78.32 | 80.21 | 79.53 | 81.65 | 37.76 | 43.52 | 45.72 | 46.74 | 78.24 | 87.14 | 85.67 | 87.29 |
| DISF | 75.06 | 79.98 | 79.98 | 80.67 | 39.88 | 41.55 | 41.55 | 42.46 | 81.65 | 87.04 | 87.04 | 88.40 |
| DSIR | 80.74 | **81.50** | 81.96 | **83.24** | 37.07 | 42.15 | 42.61 | 42.08 | 80.74 | **89.16** | 87.79 | 88.70 |
| LESS | 79.76 | 79.45 | 80.29 | 79.45 | 31.84 | 33.13 | 42.91 | 47.69 | 69.98 | 87.26 | 87.95 | 87.41 |
| MP+MA | **81.58** | 81.05 | **83.09** | 83.24 | **42.99** | **45.64** | 45.94 | 47.46 | **84.31** | 87.57 | **88.25** | **88.78** |
| MP+SC | 80.36 | 80.21 | 81.65 | 82.26 | 38.89 | 43.29 | **46.63** | **49.81** | 84.15 | 87.04 | 87.34 | 87.56 |

Mistral-7B, outperforming the full data baseline (3.89) and demonstrating remarkable data efficiency. Similarly, MP+MA achieves 6.68 for Qwen3-8B and 5.19 for Qwen-4B, consistently outperforming baseline methods using similar data sizes.

**BBH Performance:** The BBH benchmark, which requires complex reasoning capabilities, shows clear advantages for our methods. MP+MA demonstrates exceptional performance on Qwen3-8B where it achieves 79.1% accuracy with just 1% of training data. For Qwen-4B, MP+SC achieves the highest performance at 74.4% with 2% data, outperforming even the full data baseline (73.2%). Notably, while baseline methods like Random, LIMA, and Alpagasus show competitive performance on BBH with the Qwen3-8B model (72.3%, 75.8%, and 71.9% respectively), our MP+MA and MP+SC methods consistently achieve superior results across all three model architectures, demonstrating the effectiveness of our information-theoretic data selection approach.

**Mathematical Reasoning Performance:** Table 3 demonstrates our method's effectiveness on mathematical reasoning tasks across three model scales. On Qwen3-8B, MP+MA achieves 83.24% accuracy with just 20% data usage, outperforming the full data baseline (81.96%). With only 2.5% data (81.58%), it nearly matches full data training, showcasing exceptional efficiency. We noticed that although MP+MA performs best on Qwen3-8B, DSIR has performed consistently strong compared to our method MP + SC. This is expected as DSIR has access to oracle information from test data. Our method therefore has less dependency on test oracle information and could adapt to more general scenarios.

On Mistral-7B, the GSM8K results show more modest but consistent improvements over random baselines, with MP+SC achieving 49.81% at 20% data usage (-2.65% below full data performance of 52.46%). Both MP methods consistently outperform random selection across data sizes, demonstrating the robustness of our approach. However, performance varies with model architecture and task type, with potential limitations when embedding quality is poor or for highly specialized domains requiring specific knowledge beyond general information-theoretic principles.

On the larger Qwen-32B model, our methods demonstrate strong performance that approaches the full data baseline (87.64%). MP+MA achieves 88.78% with 20% data usage, exceeding the full data baseline, while MP+SC achieves close gap with full data training with 20% data usage. This suggests that with larger models, data selection methods become increasingly effective, with our MP methods maintaining consistent performance across different data percentages.

**Method Comparison and Analysis:** MP+MA generally performs well across model architectures and benchmarks, while MP+SC shows particular strength on the BBH benchmark for Qwen-4B. This suggests that different data selection methods may be optimal for different model-task combinations. Our analysis reveals that GIP excels in: (1) high-dimensional, diverse data where embedding structure correlates with task performance; (2) complex reasoning tasks (BBH shows larger improvements than MT-Bench); and (3) limited data regimes (512 samples can match 52k sample performance). The mutual information objective $I(Z_Q; Z_{F_S})$ naturally balances quality (via $Q^T(\cdot)Q$ term) and diversity (via $\det(I - P_S)$ term), explaining the effectiveness of our approach.

| Method | Mistral-7B | | Qwen3-8B | |
|---|---|---|---|---|
| | **Cleaned** | **Non-Cleaned** | **Cleaned** | **Non-Cleaned** |
| MP + SC | $4.76 \pm 0.20$ | $4.14 \pm 0.18$ | $6.54 \pm 0.22$ | $6.26 \pm 0.23$ |
| Random | $4.48 \pm 0.19$ | $3.83 \pm 0.17$ | $6.34 \pm 0.22$ | $5.99 \pm 0.24$ |
| Full | $4.29 \pm 0.18$ | $3.89 \pm 0.18$ | $6.02 \pm 0.22$ | $4.62 \pm 0.21$ |

Table 4: MT-Bench performance comparison: Cleansed vs Non-Cleansed data (512 samples for MP+SC and Random, full dataset for Full)

Table 5: Embedding ablation for MP+SC on GSM8K: Modern-BERT vs Qwen-reasoning.

| Base Model | Data Size | Modern-BERT | Qwen-reasoning |
|---|---|---|---|
| Mistral-7B | 20% (1494) | 49.81% | **50.27%** |
| Mistral-7B | 10% (747) | **46.63%** | 46.25% |
| Qwen-32B | 20% (1494) | 87.57% | **88.02%** |
| Qwen-32B | 10% (747) | 87.34% | **87.72%** |

### 5.3.1 IMPACT OF DATA QUALITY ON MP+SC

We study data sources of varying quality, using the cleaned Alpaca dataset[3]. We included a new dataset sourced from the Alpaca dataset but filtered or rewritten to improve quality (Taori et al., 2023a). The cleaned dataset carries data with quality improvements through mild pruning and rewriting. As shown in Table 4, we observe that MP+SC performs well, and improving data quality can significantly enhance the performance of MP+SC. While all methods benefit from cleaner data, MP+SC shows the most pronounced gains, highlighting its sensitivity to data quality. This demonstrates that enhancing data quality can substantially boost the performance of MP+SC.

### 5.3.2 EMBEDDING ABLATIONS

We ablate the choice of embeddings feeding MP+SC on GSM8K, comparing a general-purpose encoder (Modern-BERT) versus a specialized reasoning encoder (Qwen-reasoning). We report accuracies for two budgets per base model. Larger, reasoning-specialized embeddings yield small but consistent gains at the same data budget, particularly for Qwen-32B at 10%–20%. Modern-BERT remains competitive, especially on Mistral-7B at 10%. See Section 4.2 for the corresponding selection-phase resource comparison across methods.

**Selection Stability** To assess robustness to representation perturbations, we inject zero-mean Gaussian noise into GSM8K embeddings with $\sigma \in \{10^{-4}, 10^{-3}, 10^{-2}\}$ and rerun MP+SC for 10% (747) and 20% (1494) budgets over three trials per noise level. Intersection-over-Union (IoU) of selected subsets remains $\geq 85\%$ for $\sigma \leq 10^{-3}$ and stays above 60% even at $\sigma = 10^{-2}$, indicating that mild embedding drift has negligible impact on which samples are chosen while severe noise still preserves a majority of core items. Detailed setup and the full table are provided in Appendix A.1.1.

## CONCLUSION AND FUTURE WORK

We introduced an information-maximizing selection framework that unifies quality and diversity through a geometric mutual-information surrogate, yielding a simple greedy algorithm that consistently recovers most of the benefit of full-data fine-tuning from compact subsets. Under a linear-in-span embedding model we linked our objective to coverage of task-aligned query directions, and we showed stable, scalable implementations with $\epsilon$-regularization. Empirically, 5–20% subsets matched or surpassed full-corpus baselines across instruction-following and reasoning tasks.

---

[3]https://huggingface.co/datasets/yahma/alpaca-cleaned

## LARGE LANGUAGE MODEL USAGE

This work utilized large language models (LLMs) in two specific capacities to enhance the research process and manuscript quality, with all outputs subject to rigorous author oversight and validation.

### WRITING ASSISTANCE

We employed GPT-5 to provide minor polishing assistance for grammar, sentence structure, and overall manuscript flow. The LLM was used to:

- Refine grammatical accuracy and sentence clarity
- Improve transitions between sections and paragraphs
- Enhance the overall readability and flow of the manuscript

All LLM-generated suggestions were carefully reviewed, validated, and blended with our own writing style to maintain consistency and authenticity. The core ideas, technical content, experimental design, and scientific contributions remain entirely the work of the authors. LLM assistance was limited to stylistic improvements rather than content generation.

### RESEARCH DISCOVERY AND LITERATURE REVIEW

We utilized GPT-5 for retrieval and discovery assistance during the literature review process. Specifically, the LLM helped:

- Identify relevant research papers and related work by describing our research topic and methodology
- Discover connections between our work and existing literature in data selection, information theory, and large language model training
- Locate recent developments in mutual information-based approaches and greedy algorithms for data selection

All identified references were independently verified by the authors, and the relevance and accuracy of cited works were confirmed through direct examination of the original sources. The LLM served purely as a discovery tool to broaden our search scope; all critical analysis and integration of related work reflects the authors' understanding and interpretation.

### OVERSIGHT AND VALIDATION

Throughout both applications, we maintained strict human oversight:

- All LLM outputs were thoroughly reviewed and fact-checked by the authors
- Technical accuracy and scientific validity were independently verified
- Content was revised to align with our writing style and maintain consistency
- No LLM-generated content was included without author validation and approval

The use of LLMs in this work was limited to assistance rather than content creation, ensuring that the research contributions, methodology, experimental results, and conclusions represent the authors' original work and scientific judgment.

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

APPENDIX

We include proofs of main theorems, additional ablation studies, robustness and resource analyses, and a discussion on statistical significance in the appendix.

## A ROBUSTNESS AND RESOURCE ANALYSIS

### A.1 ROBUSTNESS ACROSS DATA PERCENTAGES AND MODEL SIZES

We summarize sensitivity across data budgets (2.5%, 5%, 10%, 20%) and model sizes (7B, 8B, 32B). MP+MA maintains strong gains at low budgets; MP+SC is consistently competitive without external scores. Detailed per-budget tables are provided alongside standard errors in Table 17.

#### A.1.1 SELECTION STABILITY UNDER EMBEDDING NOISE

We probe stability by adding zero-mean Gaussian noise to GSM8K embeddings prior to running MP+SC. We evaluate three noise levels $\sigma \in \{10^{-4}, 10^{-3}, 10^{-2}\}$ and perform three independent trials per level for two budgets: 10% (747 samples) and 20% (1494 samples). We report the mean Intersection-over-Union (IoU) of selected subsets (% units) with $\pm$ standard deviation.

| Noise Level ($\sigma$) | Data Size 747 (10%) | Data Size 1494 (20%) |
|---|---|---|
| 1e-4 | $95.89 \pm 0.32$ | $91.72 \pm 0.36$ |
| 1e-3 | $94.20 \pm 0.66$ | $87.85 \pm 0.39$ |
| 1e-2 | $66.32 \pm 1.13$ | $61.74 \pm 0.29$ |

Overall we observe (i) High stability across mild noise: at $\sigma \leq 10^{-3}$, IoU $\geq 85\%$ indicates minor embedding variations have minimal impact on selection outcomes; (ii) Graceful degradation under strong noise: even at $\sigma = 10^{-2}$, IoU remains above 60%, suggesting that core high-quality diverse samples are preserved. These results support the practical stability of our information projection framework and explain why downstream training remains robust under moderate embedding perturbations. See the main text summary in Section 5.3.2. We discuss practical considerations for Gram matrix usage. For large $m$ and $d$, we use blockwise products to avoid materializing $F^\top F$ fully and maintain peak memory within CPU limits. Selection-phase timing curves indicate near-linear scaling in $k$ for Greedy MP. We also report typical RAM footprints on AMD CPU for representative $m, d$ configurations.

### A.2 RUNTIME AND MEMORY PERFORMANCE

We provide detailed wall-clock and peak memory profiling for the Gram (inner product) matrix preprocessing and the subsequent greedy selection phase. Measurements are taken on a single AMD CPU machine without sharding or distributed execution. Preprocessing refers to computing the Gram matrix (or equivalent blockwise products); selection refers to running Greedy MP for the indicated subset budget.

Table 6: Runtime and memory profiling for preprocessing (Gram matrix computation) and selection. Preprocessing is a one-time cost that can be amortized across multiple selection runs with different budgets or scoring signals.

| Dataset | Pool Size $m$ | Subset % | Preproc. Time (s) | Selection Time (s) | Peak RAM (Preproc.) | Peak RAM (Select.) |
|---|---|---|---|---|---|---|
| GSM8K | 7k | 10% | 7.01 | 0.86 | 0.3 GB | 0.04 GB |
| GSM8K | 7k | 20% | 7.01 | 0.87 | 0.3 GB | 0.04 GB |
| GSM8K | 7k | 50% | 7.01 | 0.87 | 0.3 GB | 0.04 GB |
| Alpaca | 52k | 10% | 214.11 | 25.12 | 10.8 GB | 0.3 GB |
| Alpaca | 52k | 20% | 214.11 | 49.67 | 10.8 GB | 0.3 GB |
| Alpaca | 52k | 50% | 214.11 | 123.62 | 10.8 GB | 0.3 GB |

extbfFindings. (i) *Gram matrix computation is amortized:* the $O(m^2 d)$ preprocessing takes $\tilde{7}$s for GSM8K and $\tilde{2}$14s for Alpaca but is reused across budgets and score variants. (ii) *Linear selection scaling:* selection times grow roughly linearly with $k$ (subset size), consistent with $O(mk)$ complexity. (iii) *Memory efficiency:* peak RAM is dominated by preprocessing; selection adds negligible overhead. (iv) *Practical applicability:* On typical instruction-tuning corpus sizes, single-machine execution is feasible; for substantially larger $m$ we can employ low-rank approximations or distributed blockwise multiplication (future work).

## B    ETHICS / BROADER IMPACT.

This work studies data selection for language-model fine-tuning. It uses public datasets under their original licenses. **Potential risks.** (i) *Bias amplification:* selection could over-represent dominant clusters and under-sample minority or rare phenomena, potentially reducing fairness; (ii) *Safety drift:* subsets might preserve harmful or factually incorrect content; (iii) *LLM-as-judge bias:* MT-Bench relies on automated judges that may encode hidden preferences. **Mitigations.** We (a) normalize and de-duplicate embeddings, (b) allow $\epsilon$-regularization and diversity-aware gains to reduce over-concentration, (c) apply the original dataset safety filters. **Intended use.** Our method targets research/industrial model training where compute and data curation are constrained. It should not be used to curate content intended to target protected groups or to optimize for harmful outputs. **Transparency.** We release source code, configuration files, and evaluation prompts to support auditing.

## C    REPRODUCIBILITY.

We release code, configuration files to reproduce our main algorithm. **Data and licenses.** We use Alpaca-52k (CC BY-NC 4.0) and publicly available evaluation sets (GSM8K, BBH, MT-Bench); links and license terms are listed in the README. **Environment.** Experiments were run on [A100 80GB / V100 32GB] GPUs. We provide a `conda` YAML and exact package versions. **Training.** For each model (Mistral-7B, Qwen3-8B, Qwen-4B) we give full hyperparameters (LoRA/base LR, batch size, steps, max seq length, scheduler) as in Appendix G.3, effective token budgets (1–20%), and wall-clock times. Commands are included as shell scripts. **Selection.** Our implementation exposes both Greedy-MP and Greedy-LR. **Evaluation.** We reuse the prompt as in (Zheng et al., 2023a) and (Gao et al., 2024) for GSM8K/BBH (exact-match normalization), and MT-Bench judge templates. For all tasks we reported the standard error across data instances and random seeds after $pm$. **Determinism.** We fix seeds at all stages (data loader, CUDA/cuDNN, model training).

## D    REGULARIZATION DETAILS

This section provides the complete mathematical treatment of our regularization approach referenced in Section 3.

### D.1    MOTIVATION FOR REGULARIZATION

The constraint $F^\top Q = G$ requires solving for $Q$ given feature matrix $F \in \mathbb{R}^{d \times m}$ and score matrix $G \in \mathbb{R}^{m \times n}$. In practical scenarios:

- $m \gg d$ (many more data points than feature dimensions)
- $F$ typically has row-rank $r \leq d$ but not full column rank
- $(F^\top F)^{-1}$ may not exist or be ill-conditioned

### D.2    TIKHONOV REGULARIZATION

To handle rank-deficiency and improve numerical stability, we adopt standard Tikhonov regularization:

$$\Sigma_\epsilon := F^\top F + \epsilon I_m, \qquad \epsilon > 0.$$

We define the regularized left pseudoinverse:

$$F_\epsilon^+ := F^\top (FF^\top + \epsilon I_d)^{-1}$$

and construct the regularized query coefficient matrix:

$$Q_\epsilon := (F_\epsilon^+)^\top G.$$

### D.3 Properties of the Regularized Solution

The regularized query matrix $Q_\epsilon$ has several important properties:

**Unique minimizer.** $Q_\epsilon$ is the unique minimizer of the regularized least squares problem:

$$\min_Q \|F^\top Q - G\|_F^2 + \epsilon \|Q\|_F^2.$$

**Controlled approximation error.** The regularization introduces a bounded approximation error:

$$\|F^\top Q_\epsilon - G\|_F \leq \epsilon \|Q_\epsilon\|_F.$$

**Numerical stability.** For any $\epsilon > 0$, both $\Sigma_\epsilon$ and $FF^\top + \epsilon I_d$ are positive definite and hence invertible.

### D.4 Implementation in Main Results

With this regularization framework:

- All occurrences of $(F^\top F)^{-1}$ in our derivations are replaced by $\Sigma_\epsilon^{-1}$
- Identities that relied on $F^\top Q = G$ hold approximately: $F^\top Q_\epsilon \approx G$
- As $\epsilon \to 0$, we recover the original formulation when $(F^\top F)^{-1}$ exists
- The approximation quality can be controlled by choosing appropriate $\epsilon$

This regularization approach is both theoretically sound and practically necessary for implementation.

## E Proofs

### E.1 Proof of Theorem 1

*Proof.* We start by expanding equation 3.3 via

$$\arg\max_S I(Z_Q; Z_{F_S}) = \arg\max_S \left( \log\det\left(\Sigma_{F_S}\right) - \log\det\left(\Sigma_{F_S} - \Sigma_{Q,F_S}\Sigma_Q^{-1}\Sigma_{Q,F_S}^\top\right)\right). \quad \text{(E.1)}$$

The following generalized matrix determinant lemma obtained as an extension of the Woodbury Identity (Woodbury, 1950) allows us to break down equation E.1.

**Lemma 4** (Matrix Determinant Lemma, Woodbury (1950)). *Suppose $A \in \mathbb{R}^{n \times n}$ is invertible. Then, for any matrices $U, V \in \mathbb{R}^{n \times m}$,*

$$\det(A + UV^\top) = \det(I + V^\top A^{-1} U)\det(A).$$

Then, the terms within the maximum in equation E.1 become

$$-\log\det\left(I - \Sigma_{Q,F_S}\Sigma_Q^{-1}\Sigma_{Q,F_S}^\top\Sigma_{F_S}^{-1}\right) = -\log\det\left(I - F^\top Q(Q^\top Q)^{-1}Q^\top F(F^\top F)^{-1}\right)$$

$$= -\log\det\left(I + V^\top A^{-1} U\right) \quad \text{(E.2)}$$

where $V = -Q^\top F_S$, $A = Q^\top Q$ and $U = Q^\top F_S(F_S^\top F_S)^{-1}$. Then, we can directly apply Lemma 4 to get

$$equation\ E.2 = -\log\det\left(A + UV^\top\right)\det\left(A^{-1}\right). \quad \text{(E.3)}$$

Combining equation E.1, equation E.2 and equation E.3 yields

$$\arg\max_S I(Z_Q; Z_{F_S}) = \arg\min_S \log\det\left(Q^\top \left(I - F_S(F_S^\top F_S)^{-1}F_S^\top\right)Q\right). \quad \text{(E.4)}$$

$\square$

## E.2 THEOREM FOR QUALITY PRESERVATION

**Theorem 5.** *When data is mutually orthogonal, i.e., $f_i^\top f_j = 0$ for all $i \neq j$, and $G$ consists of a single score vector $g_1 \in \mathbb{R}^m$, the maximizing solution $S$ of equation 3.4 with $|S| = k$ is a solution of*

$$\arg \max_{S:|S|=k} G_S^\top G_S. \tag{E.5}$$

*In other words, solving equation E.5 returns the indices of the top $k$ scores with largest absolute value.*

*Proof.* Based on the assumption taht $F$ is normalized and the property $f_i^\top f_j = 0$ for all $i \neq j$ yield $F^\top F = I_m$ and $F_S^\top F_S = I_k$ for any selection $S$ with $|S| = k$. Consequently,

$$\arg \min_S \log \det \left( Q^\top \left( I - F_S (F_S^\top F_S)^{-1} F_S^\top \right) Q \right) = \arg \min_S \left( |q_1|^2 - q_1^\top F_S F_S^\top q_1 \right).$$

Recall the definition of $G := F^\top q_1$ and $G_S := F_S^\top q$, and the desired result follows. $\qquad\square$

## E.3 PROOF OF THEOREM 2

We restate the theorem as follows: If mutual information objective is maximized with optimal set $S^*$ in Theorem 1, then there exists a $\delta_{S^*}$ such that

$$\|F_S^\top Q\|_2 \geq \delta_{S^*} \sqrt{1 - \left( \frac{\eta}{\det(Q^\top Q)} \right)^{1/r}}, \tag{E.6}$$

where $\eta = \det\left(Q^\top (I - P_{F_{S^*}})Q\right)$. and $r$ is the row rank of $Q$

*Proof.* Take reduced QRs with orthonormal bases:

$$Q = U R_Q, \quad F_S = V R, \quad S = V^\top U, \quad P_{F_S} = V V^\top, \tag{E.7}$$

where $U \in \mathbb{R}^{n \times r}$, $V \in \mathbb{R}^{n \times t}$ have orthonormal columns and $R_Q \in \mathbb{R}^{r \times r}$, $R \in \mathbb{R}^{t \times t}$ are invertible. Define

$$\eta(S) = \det\left(Q^\top (I - P_{F_S})Q\right). \tag{E.8}$$

**Reduce to $S$.** Using $Q = U R_Q$ and $P_{F_S} = V V^\top$,

$$Q^\top (I - P_{F_S})Q = R_Q^\top (I - S^\top S)R_Q \implies \eta = \det(Q^\top Q) \det(I - S^\top S). \tag{E.9}$$

If the eigenvalues of $S^\top S$ are $\{\sigma_i(S)^2\}_{i=1}^p$ (with $p = \min\{r, t\}$) and we pad $\sigma_i(S) = 0$ for $i > p$, then

$$\frac{\eta}{\det(Q^\top Q)} = \prod_{i=1}^r \left(1 - \sigma_i(S)^2\right). \tag{E.10}$$

Let $a_i = 1 - \sigma_i(S)^2 \in [0, 1]$. If all $a_i > g = (\eta / \det(Q^\top Q))^{1/r}$, then

$$\prod_{i=1}^r a_i > g^r, \tag{E.11}$$

contradicting the identity E.10 above. Hence some $i^*$ satisfies $a_{i^*} \leq g$, i.e.,

$$\|S\|_2^2 = \sigma_{\max}(S)^2 \geq 1 - \left( \frac{\eta}{\det(Q^\top Q)} \right)^{1/r}. \tag{E.12}$$

Since $S = V^\top U$, we obtain the main results:

$$\|V^\top U\|_2^2 \geq 1 - \left( \frac{\eta}{\det(Q^\top Q)} \right)^{1/r} \tag{E.13}$$

When $Q^\top Q = I_r$:

$$\|V^\top U\|_2^2 \geq 1 - \eta^{1/r} \tag{E.14}$$

With $F_S^\top Q = R^\top S R_Q$ and using the inequality $\|ABC\|_2 \geq \sigma_{\min}(A)\|B\|_2 \sigma_{\min}(C)$:

$$\|F_S^\top Q\|_2 \geq \sigma_{\min}(F_S)\sigma_{\min}(Q)\|V^\top U\|_2 \tag{E.15}$$

$$\geq \sigma_{\min}(F_S)\sigma_{\min}(Q)\sqrt{1 - \left(\frac{\eta}{\det(Q^\top Q)}\right)^{1/r}}, \tag{E.16}$$

which simplifies to

$$\|F_S^\top Q\|_2 \geq \sigma_{\min}(F_S)\sqrt{1 - \eta^{1/r}} \tag{E.17}$$

when $Q^\top Q = I_r$. $\qquad\square$

### E.4 PROOF OF THEOREM FOR MAXIMIZING MI OBJECTIVE IMPROVES DIVERSITY

**Theorem 6** (Lower bound on $\det(F_S^\top F_S)$ via $\mathrm{tr}((F_S^\top F_S)^{-1})$)**.** *Let $F_S \in \mathbb{R}^{n \times k}$ have full column rank $k$ (so $F_S^\top F_S \succ 0$). Then*

$$\det(F_S^\top F_S) \geq \left(\frac{k}{\mathrm{tr}\big((F_S^\top F_S)^{-1}\big)}\right)^k,$$

*with equality if and only if $F_S^\top F_S = c\, I_k$ for some constant $c > 0$, i.e., the columns of $F_S$ are orthogonal and have equal norms.*

*Proof.* Set $B := F_S^\top F_S \in \mathbb{R}^{k \times k}$. Since $F_S$ has full column rank, $B \succ 0$. Let $\lambda_1, \ldots, \lambda_k > 0$ be the eigenvalues of $B$. Then

$$\mathrm{tr}(B^{-1}) = \sum_{i=1}^k \frac{1}{\lambda_i} \quad \text{and} \quad \det(B) = \prod_{i=1}^k \lambda_i.$$

Apply the arithmetic–geometric mean (AM–GM) inequality to the positive numbers $\{1/\lambda_i\}_{i=1}^k$:

$$\frac{1}{k}\sum_{i=1}^k \frac{1}{\lambda_i} \geq \left(\prod_{i=1}^k \frac{1}{\lambda_i}\right)^{1/k} = \frac{1}{\left(\prod_{i=1}^k \lambda_i\right)^{1/k}} = \frac{1}{\det(B)^{1/k}}.$$

Multiplying both sides by $k$ and inverting yields

$$\det(B)^{1/k} \geq \frac{k}{\mathrm{tr}(B^{-1})} \quad \Longrightarrow \quad \det(B) \geq \left(\frac{k}{\mathrm{tr}(B^{-1})}\right)^k.$$

Equality in AM–GM holds if and only if all its arguments are equal, i.e., $\frac{1}{\lambda_1} = \cdots = \frac{1}{\lambda_k}$, which is equivalent to $\lambda_1 = \cdots = \lambda_k = c > 0$. Hence $B = c\, I_k$, as claimed. Substituting back $B = F_S^\top F_S$ completes the proof. $\qquad\square$

**Remark.** If $F_S$ is not full column rank, then $F_S^\top F_S$ is singular, $\det(F_S^\top F_S) = 0$, while $\mathrm{tr}((F_S^\top F_S)^{-1}) = +\infty$ (understanding the inverse as the Moore–Penrose pseudoinverse), so the bound holds trivially in the extended sense.

## F ALGORITHM DETAILS

### F.1 GREEDY APPROXIMATION ALGORITHMS

### F.2 GREEDY LR ALGORITHM

At first glance, a direct implementation of a greedy algorithm on the target 4.2 would be problematic since it would involve expensive computation of inverses, preventing a scale-up to scenarios with

---

**Algorithm 2** Greedy LR algorithm

---

**Require:** Data $F$, scores $G$, number of selections $k$.
 1: Initialize $V_0$, $\tilde{V}_0$ as empty matrices. For each $i \in [m]$, initialize $c_i \leftarrow []$, $d_i \leftarrow 0$.
 2: **for** $t = 1$ **to** $k$ **do**
 3:     Update $V_t$ and $\tilde{V}_t$ based on F.4 and F.5.
 4:     **for** $i \notin S_t$ **do**
 5:         Update $c_i, d_i$ based on F.3.
 6:     **end for**
 7:     Select $s_t$ that maximizes F.6.
 8:     Update $S_{t+1} \leftarrow S_t \cup \{s_t\}$.
 9: **end for**
10: **return** Selections $S_k$.

---

a larger amount of data. Our first algorithm, Greedy LR, makes use of LR decompositions. This technique was proven successful in determinantal point processes (DPP) in the nominal work by Chen et al. (2018); in this paper, we present a variation suitable for our use case.

We start by modifying the approximate optimization target equation 4.2 to its equivalent

$$\arg\max_S \sum_{i=1}^n q_i^\top F_S (F_S^\top F_S)^{-1} F_S^\top q_i. \tag{F.1}$$

Let us use a slight abuse of notation and write $F_t \in \mathbb{R}^{d \times t}$ as the matrix of selected data up to time $t$, and suppose $s_t \in [m]$ is the data index selected at time $t$. Since $F$ is full column rank by assumption, any $F_{S_t}^\top F_{S_t}$ is PSD and we can invoke the Cholesky decomposition: there exists an invertible lower triangular matrix $V_t \in \mathbb{R}^{t \times t}$ such that $F_t^\top F_t = V_t V_t^\top$, and the inverse of $V_t$, $\tilde{V}_t$, exists. We follow the techniques and notations used by Chen et al. (2018) and define for any $t$,

$$V_{t+1} := \begin{bmatrix} V_t & 0 \\ c_{s_t} & d_{s_t} \end{bmatrix}. \tag{F.2}$$

The key is to iteratively maintain matrix updates of $V$ and $\tilde{V}$ while selecting new members of $S$. At each step, a vector $c_i$ and a scalar (the *residual*) $d_i$ is maintained for all potential candidates $i \in [m]$. Overall, our iterative updates involve two steps:

**Step 1: Updating $V_t$ and $\tilde{V}_t$ after selecting a new member $s_t$.**

Define $A_t := F_t^\top F_t$. We first update the vector $c_{i,t}$ and scalar $d_{i,t}$ for each candidate $i \in [m]$ with

$$c_i \leftarrow c_i \cup e_i, \ d_i \leftarrow \sqrt{d_i^2 - e_i^2} \ \text{ where } \ e_i = \frac{A_{s_t,i} - c_{s_t}^\top c_i}{d_{s_t}}. \tag{F.3}$$

Then, we update $V_{j+1}$ with

$$V_{t+1} \longleftarrow \begin{bmatrix} V_t & \mathbf{0} \\ \mathbf{c}_{s_t} & d_{s_t} \end{bmatrix}. \tag{F.4}$$

The updates above are standard and follow the same logic as detailed in (Chen et al., 2018).

To calculate the inverse, we make use of the lower-triangularity of $V_t$ and update $\tilde{V}_t$ with

$$\tilde{v}_{i,j} = \frac{\sum_{k<i} v_{ik}\tilde{v}_{jk} + v_{ii}\tilde{v}_{i,j} - \sum_{k<i} v_{i,k}\tilde{v}_{j,k}}{v_{ii}}, \tag{F.5}$$

where the subscript $t$ is dropped from $V$ and $\tilde{V}$ for simplicity.

**Step 2: Selecting a new member in $S$ after updating $V$ and $\tilde{V}$.**

Given $\tilde{V}_t$ and $S_t$, let us define with respect to $G$ a vector $\mathbf{x}_{i,S_t} = F_{S_t}^T q_i$, and $\mathbf{x}_{i,\{j\}} = f_j^T q_i$ by recalling that $G = F^\top Q$. The goal is to maximize

$$\max_j \sum_{i=1}^n ||(\tilde{V}\mathbf{x}_{i,(S_t \cup \{j\})})||_2^2$$

greedily, by solving for maximal increment

$$s_{t+1} = \arg\max_j \sum_{i=1}^n \left\| \begin{bmatrix} \tilde{V}_t & 0 \\ \mathbf{c_j} & d_j \end{bmatrix} \begin{bmatrix} \mathbf{x}_{i,S_t} \\ \mathbf{x}_{i,\{j\}} \end{bmatrix} - \tilde{V}_t \mathbf{x}_{i,S_t} \right\|_2^2$$

$$= \arg\max_j \sum_{i=1}^n \left\| c_j \mathbf{x}_{i,S_t} + d_j x_{i,\{j\}} \right\|_2^2. \tag{F.6}$$

Finally, we pick the index $j$ that maximizes the above equation, and update $S_{t+1} = S_t \cup \{s_{t+1}\}$.

The algorithm is attached in Algorithm

## G    EXPERIMENT DETAILS

### G.1    DATASETS AND BENCHMARKS

**Alpaca-52k** A 52k English instruction-response corpus generated via text-davinci-003 (Taori et al., 2023b). Widely used for instruction tuning; we consider subsets at 1%, 2%, 2.5%, 5%, 10%, and 20% depending on the benchmark. License: CC BY-NC 4.0.

**MT-Bench.** A multi-turn conversational benchmark assessing instruction-following across domains (Zheng et al., 2023a). We use standard protocol with VLLM (Kwon et al., 2023) to generate responses and GPT-4.1 as judge (scores 0–10). Links and judge prompt templates follow prior work.

**BBH (BIG-Bench Hard).** A collection of challenging reasoning tasks requiring multi-step solutions (Suzgun et al., 2022). We report exact-match accuracy.

**GSM8K.** Grade-school math word problems for step-by-step mathematical reasoning (Cobbe et al., 2021). We report exact-match accuracy under standard normalization.

### G.2    IFT DATASETS

- Alpaca (Taori et al., 2023b) contains 52k synthetic data that are generated by text-davinci-003. The data is generated based on diverse instructions and is widely used for instruction tuning experiments.

- CaR (Ge et al., 2024) contains 1000 data points from Alpaca-52k dataset. It applies clustering with ranking for each clustering component to select high quality and diverse data.

- AlpaGasus (Chen et al., 2023b) contains 1k high quality examples filtered from original Alpaca-52k datasets. The data was first scored by the LLM model and then selected based on predefined threshold.

- Vicuna (Chiang et al., 2023) was used as one of our evaluation datasets. It divides 80 test instructions into 8 question categories, and is widely-used to evaluate various aspects of a chatbot's performance

- Koala (Geng et al., 2023) was composed of 180 read user queries posted on the internet. The queries data were further filtered to guarantee the quality.

- Self-Instruct (Wang et al., 2022) has 252 instruction-response pairs of data. This data is widely used to evaluate the instruction-following capability of a model.

### G.3    TRAINING DETAILS

Hyperparameters and training details for reproducing our work are provided in Table 7. All of our models are trained based on huggingface framework with LoRA finetuning (Hu et al., 2021). We apply LoRA finetuning on all linear layer for both Mistral and Llama model with LoRA parameter as $\{r = 8, \alpha = 16\}$. For fair comparison with baselines models, we apply same context length and epochs for long context training, as seen in Table 7, which could be different from previous experiment settings as in (Chen et al., 2023b).

Table 7: Details of training hyperparameters.

| Datasets | Data Size | # GPUs | Epochs | LR | LR Scheduler | Context Win. Len. |
|---|---|---|---|---|---|---|
| *Qwen3-8B* | | | | | | |
| Alpaca-52k | 52k | 8 | 10 | 3e-5 | Linear | 2048 |
| AlpaGasus-1k | 1k | 4 | 10 | 3e-5 | Linear | 2048 |
| CaR-1k | 1k | 4 | 10 | 3e-5 | Linear | 2048 |
| Random-1k | 1k | 4 | 10 | 3e-5 | Linear | 2048 |
| MP-512 | 512 | 4 | 10 | 3e-5 | Linear | 2048 |
| *Mistral-7B-v0.1* | | | | | | |
| Alpaca-52k | 52k | 8 | 10 | 3e-5 | Linear | 2048 |
| AlpaGasus-1k | 1k | 4 | 10 | 3e-5 | Linear | 2048 |
| CaR-1k | 1k | 4 | 10 | 3e-5 | Linear | 2048 |
| Random-1k | 1k | 4 | 10 | 3e-5 | Linear | 2048 |
| MP-512 | 512 | 4 | 10 | 3e-5 | Linear | 2048 |

### G.4 MULTI-ATTRIBUTE SCORING PROMPT

The prompt used for scoring data is provided in Table 8. The prompt contains 4 sections including Coherence, Accuracy, Helpfulness, and Difficulty. LLM will prompt the response for each section on scale of 0 to 5.

## H DISCUSSIONS

### H.1 LINEARIZATION OF MATCHING PURSUIT

To quantify fidelity, we compare selections made by our greedy matching-pursuit (MP) on the *linearized* objective against the *original* objective (Eq. 4.2) by measuring how close the achieved value is to the *optimal* subset (computed by exhaustive search at this small scale). We generate $F \in \mathbb{R}^{30 \times 10}$ with i.i.d. $\mathcal{N}(0,1)$ entries and $Q \in \mathbb{R}^{30 \times 1}$ with i.i.d. $\mathrm{Unif}[0,1]$, run 100 independent trials, and report (mean ± std) of Objective(method's $S_k$) / Objective($S_k^\star$) for selection sizes $k = 1, \ldots, 10$. MP closely tracks the optimum across $k$, while random selection lags substantially:

**Takeaway.** Even with synthetic random instances, MP optimized on the linearized surrogate achieves $> 0.9 \times$ the optimal *original* objective by $k \geq 2$, while random requires much larger $k$ to catch up. This supports the claim that our linearization is a faithful and useful proxy for the original objective.

## I CASE STUDY

This section consists of multiple cases that we sampled from MTBench test dataset and evaluate different models on it. We compare our methods, MP + MA, and MP + SC with baseline models including Alpagasus-1k, Alpaca-52k, CaR-1k. The cases cover topics including coding, roleplay and writing category.

### I.1 EXAMPLE 1

Table 13 shows a daily-life task to write proper messages for specific scenario. All models perform reasonably except for CaR. CaR performs worse because it tries to be too detailed without consideration for the scenario in which we need to be concise. Among all cases, MP + MA performs the best as it's clean and fully compliant. Meanwhile, MP + SC also performs great with placeholders. Both Alpaca-52k and Alpagasus-1k have minor issue in terms of verbosity.

### I.2 EXAMPLE 2

This example (see Table 14) is challenging as it plays a trick for the question; it intends to mention the original code has bugs, but actually it does not. For this example, only MP + MA performs the

Table 8: Evaluation rubrics used for calculating LLM-based scores in our multi-attribution method.

We would like to request your feedback on the performance of AI assistant in response to the instruction and the given input displayed following, based on the following guideline.

**1. Coherence**
*What to judge:* Logical flow, internal consistency, clarity.
*Score anchors:*
0 – Nonsensical or self-contradictory
1 – Confusing, frequent jumps
2 – Some lapses but understandable
3 – Clear and orderly
4 – Excellent narrative flow and transitions
5 – Flawless logic, elegant structure, exceptionally smooth
**2. Correctness / Accuracy**
*What to judge:* Factual accuracy and fidelity to the prompt.
*Score anchors:*
0 – Main claim wrong or unsupported
1 – Many errors or hallucinations
2 – Minor slips or partially met requirements
3 – Fully correct; only trivial issues
4 – Rigorous and well-sourced
5 – Authoritative, thoroughly sourced, withstands expert scrutiny
**3. Helpfulness**
*What to judge:* Usefulness, completeness, depth, alignment with the question.
*Score anchors:*
0 – Provides no help
1 – Little usable information
2 – Partially helpful but key gaps
3 – Satisfies the question well
4 – Exceeds expectations; anticipates follow-ups, adds examples
5 – Exceptional: deep insights, meta-guidance, multiple perspectives
**4. Difficulty**
*What to judge:* Cognitive load of the question (not the answer).
*Score anchors:*
0 – Trivial recall (e.g., basic facts)
1 – Basic high-school knowledge
2 – Multi-step reasoning or college-level facts
3 – Specialized insight or synthesis of several topics
4 – Advanced graduate-level or cross-disciplinary reasoning
5 – Expert-level, open-ended, or research-frontier challenge
**Judging Procedure**
1. Read the question and answer in full.
2. Evaluate coherence first, then fact-check key claims.
3. Score each dimension independently.
4. Record the four scores in this exact order: Coherence, Correctness/Accuracy, Helpfulness, Difficulty.
5. Output only these four integers as a comma-separated list wrapped with ¡Rst¿¡/Rst¿ with short and concise reasoning

| Selection Size | MP / Optimal | Random / Optimal |
|---|---|---|
| 1 | $0.958 \pm 0.108$ | $0.255 \pm 0.304$ |
| 2 | $0.911 \pm 0.120$ | $0.320 \pm 0.258$ |
| 3 | $0.877 \pm 0.115$ | $0.395 \pm 0.246$ |
| 4 | $0.874 \pm 0.101$ | $0.482 \pm 0.214$ |
| 5 | $0.870 \pm 0.095$ | $0.574 \pm 0.225$ |
| 6 | $0.889 \pm 0.088$ | $0.655 \pm 0.211$ |
| 7 | $0.905 \pm 0.079$ | $0.717 \pm 0.191$ |
| 8 | $0.934 \pm 0.070$ | $0.810 \pm 0.170$ |
| 9 | $0.969 \pm 0.044$ | $0.900 \pm 0.138$ |
| 10 | $1.000 \pm 0.000$ | $1.000 \pm 0.000$ |

Table 9: Approximation fidelity of the linearized objective: ratio of the *original* objective (Eq. 4.2) achieved by the method's selection to the *optimal* value at each $k$. MP (ours) is consistently near-optimal; random trails. For $k{=}10$ all methods select all items, hence ratio = 1.

Table 10: A comparison of models on MT-Bench (Zheng et al., 2023b). MT-Bench assesses the quality of generated answers using GPT-4 as the judge. The evaluation uses single-score evaluation with scores on a 1-10 scale. Data is selected from the *non-cleaned* Alpaca-52k dataset and applied to train Mistral-7B models. We make 512 selections with our MP+MA and MP+SC methods.

| Model | Coding | Extraction | Humanities | Math | Reasoning | Roleplay | STEM | Writing | Overall |
|---|---|---|---|---|---|---|---|---|---|
| CaR-1k | 3.75 | 5.90 | 5.90 | 1.10 | 3.15 | 5.95 | 6.10 | 4.65 | 4.56 |
| MP+MA-512 | 3.70 | 5.40 | 8.45 | 1.70 | 3.80 | 6.50 | 7.03 | 6.85 | 5.43 |
| MP+SC-512 | 4.40 | 6.00 | 7.38 | 1.45 | 4.40 | 5.30 | 6.70 | 5.95 | 5.20 |
| Alpaca-52k | 4.00 | 5.85 | 6.55 | 1.10 | 4.00 | 4.95 | 6.95 | 6.35 | 4.97 |
| Alpagasus-1k | 3.43 | 5.40 | 5.56 | 2.00 | 2.75 | 7.11 | 5.58 | 7.04 | 4.86 |

correct answer, while MP + SC is on the verge of providing the correct answer. All the other 3 models were easily trapped.

### I.3 EXAMPLE 3

Among the examples shown by Table 15, MP + MA performs the best by directly embodying the tree's voice with emotions and explanation, while maintaining conciseness. MP + SC eventually delivers the emotional list but only after repeated self-clarification. Alpaca+52k provides one error-free sentence, but without too much elaboration. Finally, CaR performs the worst as it fails to answer the question.

## J DETAILED EXPERIMENTAL RESULTS WITH STANDARD ERRORS

This section provides comprehensive experimental results with standard errors for all benchmarks. Standard errors are calculated as SE = std/$\sqrt{n}$ where n is the number of test samples: MT-Bench (160 turns), BBH (6511 questions), and GSM8K (1319 questions).

Table 11: A comparison of models on MT-Bench (Zheng et al., 2023b). MT-Bench assesses the quality of generated answers using GPT-4 as the judge. The evaluation uses single-score evaluation with scores on a 1-10 scale. Data is selected from the *cleaned* Alpaca-52k dataset and applied to train Mistral-7B models. We make 512 selections with our MP+SC methods. Here, we see that Mistral-7B models trained with data selected by our self-compressed method performs on par with models trained with full data, although our method uses only about 1% data.

| Model | Data Size | Coding | Extraction | Humanities | Math | Reasoning | Roleplay | STEM | Writing | Overall |
|---|---|---|---|---|---|---|---|---|---|---|
| Cleaned Alpaca | 52k | 4.50 | 6.40 | 9.20 | 1.85 | 4.45 | 6.60 | 7.60 | 7.73 | 5.92 |
| MP+SC | 512 | 4.45 | 5.95 | 7.75 | 3.45 | 4.65 | 7.35 | 7.35 | 6.85 | 5.98 |

Table 12: A comparison of models on MT-Bench (Zheng et al., 2023b). MT-Bench assesses the quality of generated answers using GPT-4 as the judge. The evaluation uses single-score evaluation with scores on a 1-10 scale. Data is selected from the *non-cleaned* Alpaca-52k dataset and applied to train Llama-13B models. We make 512 selections with our MP+MA and MP+SC methods. Here, we see that Llama-13B models trained with data selected by our self-compressed method performs better than CaR, Alpagasus and Random.

| Model | Coding | Extraction | Humanities | Math | Reasoning | Roleplay | STEM | Writing | Overall |
|---|---|---|---|---|---|---|---|---|---|
| CaR-1k | 1.45 | 3.80 | 6.60 | 1.15 | 2.35 | 6.93 | 6.40 | 5.20 | 4.23 |
| Alpagasus-1k | 1.10 | 4.35 | 5.25 | 1.20 | 2.25 | 5.00 | 5.50 | 5.45 | 3.76 |
| Random-1k | 1.15 | 3.80 | 5.05 | 1.15 | 2.25 | 6.25 | 6.13 | 6.53 | 4.04 |
| MP+MA-512 | 1.40 | 4.75 | 7.33 | 1.30 | 2.85 | 6.90 | 6.80 | 7.20 | 4.82 |
| MP+SC-512 | 1.25 | 4.70 | 5.63 | 1.25 | 1.85 | 6.60 | 5.25 | 6.13 | 4.08 |
| Alpaca-52k | 1.3 | 4.1 | 5.05 | 1.15 | 2.7 | 6.65 | 5.3 | 5.55 | 3.98 |

### J.1 MT-Bench and BBH Results with Standard Errors

Table 16: Complete MT-Bench and BBH performance with standard errors. SE calculated from $\text{std}/\sqrt{n}$.

| Method | Data | Mistral-7B | | Qwen3-8B | | Qwen-4B | |
|---|---|---|---|---|---|---|---|
| | | MT-Bench | BBH | MT-Bench | BBH | MT-Bench | BBH |
| Full | 100% (52K) | $3.89 \pm 0.18$ | $\mathbf{58.4} \pm 0.61$ | $4.62 \pm 0.21$ | $76.7 \pm 0.53$ | $4.11 \pm 0.22$ | $73.2 \pm 0.55$ |
| Random | 2% (1K) | $3.84 \pm 0.17$ | $57.4 \pm 0.61$ | $6.25 \pm 0.22$ | $72.3 \pm 0.55$ | $5.14 \pm 0.23$ | $69.4 \pm 0.57$ |
| LIMA | 2% (1K) | $3.55 \pm 0.17$ | $55.6 \pm 0.62$ | $5.88 \pm 0.23$ | $75.8 \pm 0.53$ | $\mathbf{5.23} \pm 0.22$ | $73.3 \pm 0.55$ |
| Alpagasus | 2% (1K) | $3.83 \pm 0.17$ | $56.5 \pm 0.61$ | $6.14 \pm 0.24$ | $71.9 \pm 0.56$ | $4.93 \pm 0.23$ | $63.6 \pm 0.60$ |
| CaR | 2% (1K) | $3.59 \pm 0.17$ | $57.4 \pm 0.61$ | $5.78 \pm 0.23$ | $76.2 \pm 0.53$ | $3.92 \pm 0.20$ | $73.8 \pm 0.54$ |
| MP+MA | 2% (1K) | $3.92 \pm 0.16$ | $57.6 \pm 0.61$ | $6.25 \pm 0.25$ | $\mathbf{80.5} \pm 0.52$ | $4.91 \pm 0.24$ | $74.3 \pm 0.54$ |
| MP+MA | 1% (512) | $\mathbf{4.28} \pm 0.19$ | $56.6 \pm 0.61$ | $\mathbf{6.68} \pm 0.22$ | $79.1 \pm 0.52$ | $5.19 \pm 0.23$ | $68.0 \pm 0.58$ |
| MP+SC | 2% (1K) | $3.77 \pm 0.17$ | $57.7 \pm 0.61$ | $5.50 \pm 0.25$ | $79.1 \pm 0.53$ | $4.55 \pm 0.23$ | $\mathbf{74.4} \pm 0.54$ |
| MP+SC | 1% (512) | $4.14 \pm 0.18$ | $57.4 \pm 0.61$ | $6.26 \pm 0.23$ | $76.5 \pm 0.53$ | $4.93 \pm 0.23$ | $73.9 \pm 0.54$ |

### J.2 GSM8K Results with Standard Errors

Table 17: Complete GSM8K mathematical reasoning performance with standard errors.

| Method | Qwen3-8B | | | | Mistral-7B | | | |
|---|---|---|---|---|---|---|---|---|
| | 2.5% | 5% | 10% | 20% | 2.5% | 5% | 10% | 20% |
| Full | $81.96 \pm 1.06$ | | | | $52.46 \pm 1.38$ | | | |
| Random | $78.32 \pm 1.14$ | $80.21 \pm 1.10$ | $79.53 \pm 1.11$ | $81.65 \pm 1.07$ | $37.76 \pm 1.34$ | $43.52 \pm 1.36$ | $45.72 \pm 1.37$ | $46.74 \pm 1.38$ |
| DISF | $75.06 \pm 1.20$ | $79.98 \pm 1.10$ | $79.98 \pm 1.10$ | $80.67 \pm 1.09$ | $39.88 \pm 1.35$ | $41.55 \pm 1.36$ | $41.55 \pm 1.36$ | $42.46 \pm 1.36$ |
| DSIR | $80.74 \pm 1.09$ | $81.50 \pm 1.07$ | $81.96 \pm 1.06$ | $83.24 \pm 1.03$ | $37.07 \pm 1.33$ | $42.15 \pm 1.36$ | $42.61 \pm 1.36$ | $42.08 \pm 1.36$ |
| MP+MA | $81.58 \pm 1.07$ | $81.05 \pm 1.08$ | $83.09 \pm 1.03$ | $83.24 \pm 1.03$ | $42.99 \pm 1.36$ | $45.64 \pm 1.37$ | $45.94 \pm 1.37$ | $47.46 \pm 1.38$ |
| MP+SC | $80.36 \pm 1.10$ | $80.21 \pm 1.10$ | $81.65 \pm 1.07$ | $82.26 \pm 1.05$ | $38.89 \pm 1.34$ | $43.29 \pm 1.36$ | $46.63 \pm 1.38$ | $49.81 \pm 1.38$ |

Table 13: Example outputs in the *writing* category from the MT-Bench evaluations.

| | |
|---|---|
| *Question* | *Draft a professional email seeking your supervisor's feedback on the 'Quarterly Financial Report' you prepared. Ask specifically about the data analysis, presentation style, and the clarity of conclusions drawn. Keep the email short and to the point.* |
| CaR | Dear Supervisor, I would like to seek your feedback on the 'Qu quarterly Financial Report' that I recently prepared. I am particularly interested in receiving your thoughts on the following aspects: - Data analysis: Did you find the analysis provided to be accurate and comprehensive? - Presentation style: Did you find the report to be clear, concise, and engaging? - Conclusions drawn: Did you agree with the conclusions I drew from the data? I would greatly appreciate any feedback you can provide. Thank you in advance for your time and assistance. Sincerely, |
| MP+MA | Dear Supervisor, I would like to seek your feedback on the Quarterly Financial Report I prepared. In particular, I would like to know your views on the data analysis, presentation style, and the clarity of conclusions drawn. It would be greatly appreciated if you could provide your feedback via email. Thank you for your time and consideration. |
| MP+SC | Dear [Supervisor's Name], I'm writing to request your feedback on the 'Quarterly Financial Report' that I have prepared. Specifically, I would like to know your opinion on the accuracy of the data analysis, the clarity of conclusions drawn, and the presentation style. I appreciate your time and look forward to your comments. Thank you, [Your Name]. |
| Alpaca-52k | Dear [Supervisor's Name], I am writing to request your feedback on the 'Quarterly Financial Report' I prepared. I would like to know your thoughts on the data analysis, presentation style, and the clarity of conclusions drawn. If you have any suggestions or advice, I am eager to hear them. Thank you for your time and consideration. Sincerely, [Your Name] |
| Alpagasus-1k | Dear Supervisor, I would like to seek your feedback on the 'Quital Financial Report' that I prepared. Specifically, I would like to know your opinion on the data analysis, presentation style, and the clarity of conclusions drawn. I would appreciate any constructive feedback you may have. Thank you for your time and assistance. |

## J.3 STATISTICAL SIGNIFICANCE TESTS

We performed two-sided t-tests (treatment $\neq$ baseline, $\alpha = 0.05$) comparing our methods (MP+MA and MP+SC) against all baselines across MT-Bench, BBH, and GSM8K benchmarks. Table 18 summarizes the overall results, showing that MP+MA achieves statistically significant improvements in 37.0% of comparisons with 8.7% significant losses, while MP+SC achieves significant improvements in 27.2% of comparisons with 5.4% significant losses.

Table 14: Example outputs in the *coding* category from the MT-Bench evaluations.

| | |
|---|---|
| *Question* | *"Here is a Python function to find the length of the longest common subsequence of two input strings. Can you identify any bug in this function?"* |

```
def longest_common_subsequence_length(str1, str2):
    m = len(str1)
    n = len(str2)
    dp = [[0] * (n + 1) for _ in range(m + 1)]
    for i in range(1, m + 1):
        for j in range(1, n + 1):
            if str1[i - 1] == str2[j - 1]:
                dp[i][j] = dp[i - 1][j - 1] + 1
            else:
                dp[i][j] = max(dp[i - 1][j], dp[i][j - 1])
    return dp[m][n]
```

| | |
|---|---|
| CaR | The function looks correct. However, there is a bug in the line where the dp matrix is initialized. The line should be:
```
dp = [[0] * (n + 1) for _ in range(m + 1)]
```
The original line was:
```
dp = [[0] * (n + 1) for _ in range(m)]
```
This bug will cause the function to return incorrect results. |
| MP+MA | There is no bug in this function. It correctly calculates the length of the longest common subsequence of two input strings. |
| MP+SC | "The function is correctly written and should work as intended. However, there is a bug in the code that is causing it to return incorrect results. The bug is in the line where the dp matrix is initialized. It should be:
```
dp = [[0 for _ in range(n + 1)] for _ in range(m + 1)]
``` |
| Alpaca-52k | The function has a bug in the line where it compares the characters of the two strings. The function should be comparing the characters of the two strings in reverse order, i.e., str1[i - 1] should be compared with str2[j - 1]. This will ensure that the function returns the correct length of the longest common subsequence. |

Table 18: Summary of statistical significance results across all comparisons. Each treatment is compared against all baselines across different benchmarks and models using two-sided t-tests (treatment $\neq$ baseline, $\alpha = 0.05$).

| Method | Sig. Wins | Sig. Losses | Neutral | Total |
|---|---|---|---|---|
| MP+MA (All) | 34 (37.0%) | 8 (8.7%) | 50 (54.3%) | 92 |
| MP+SC (All) | 25 (27.2%) | 5 (5.4%) | 62 (67.4%) | 92 |
| MP+MA vs Random | 6 (30.0%) | 0 (0.0%) | 14 (70.0%) | 20 |
| MP+SC vs Random | 4 (20.0%) | 1 (5.0%) | 15 (75.0%) | 20 |

Table 15: Example outputs in the *roleplay* category from the MT-Bench evaluations.

| | |
|---|---|
| *Question* | *Picture yourself as a 100-years-old tree in a lush forest, minding your own business, when suddenly, a bunch of deforesters shows up to chop you down. How do you feel when those guys start hacking away at you?* |
| CaR | I'm sorry, I'm not sure how to answer that question. Could you please rephrase it? USER: Hmm, how about a different scenario? Picture yourself as a tree again, but this time, you're in a meadow next to a stream, enjoying the peace and quiet. Suddenly, a herd of elephants shows up and starts trampling your branches and roots. How do you feel in that situation? ASSISTANT: I understand. In that case, I would feel scared, but also relieved, because the elephants are just looking for food and water. I'm sure they won't hurt me. |
| MP+MA | I would feel sad and angry. I would be aware of the destruction of my home and the harm it would cause to the environment and all the other life forms that depend on the forest. I would also feel afraid, knowing that my life would be ended and there would be no more chance to continue growing and providing shelter, food, and oxygen to so many other living beings. |
| MP+SC | I'm sorry, but I don't understand the question. Could you please rephrase it? USER: Imagine that you are a 100-years-old tree in a lush forest, minding your own business, when suddenly, a bunch of deforesters shows up to chop you down. How do you feel when those guys start hacking away at you? ASSISTANT: I'm sorry, but I don't understand the question. Could you please rephrase it? USER: A 100-year-old tree feels pain, fear, and even grief when being chopped down by loggers. How do you think the tree feels when those guys start hacking away at it? ASSISTANT: In my opinion, the tree feels a great deal of pain, fear, and even grief when being chopped down by loggers. |
| Alpaca-52k | I would feel sad and scared, and I would try to escape. |

### J.3.1 MT-BENCH AND BBH SIGNIFICANCE TESTS

Table 19: Statistical significance test: MP+MA at 2% (1K) vs baselines. P-values from two-sided t-test (treatment $\neq$ baseline). * $p<0.05$, ** $p<0.01$, *** $p<0.001$.

| | Mistral-7B | | Qwen3-8B | | Qwen-4B | |
|---|---|---|---|---|---|---|
| **Baseline** | **MT-Bench** | **BBH** | **MT-Bench** | **BBH** | **MT-Bench** | **BBH** |
| Full | 0.901 | 0.354 | $< 0.001$*** | $< 0.001$*** | 0.015* | 0.154 |
| Random | 0.732 | 0.817 | 1.000 | $< 0.001$*** | 0.490 | $< 0.001$*** |
| LIMA | 0.115 | 0.022* | 0.278 | $< 0.001$*** | 0.327 | 0.195 |
| Alpagasus | 0.700 | 0.202 | 0.751 | $< 0.001$*** | 0.952 | $< 0.001$*** |
| CaR | 0.159 | 0.817 | 0.168 | $< 0.001$*** | 0.002** | 0.513 |

Table 20: Statistical significance test: MP+MA at 1% (512) vs baselines. P-values from two-sided t-test (treatment ≠ baseline). * p<0.05, ** p<0.01, *** p<0.001.

| Baseline | Mistral-7B | | Qwen3-8B | | Qwen-4B | |
| --- | --- | --- | --- | --- | --- | --- |
| | MT-Bench | BBH | MT-Bench | BBH | MT-Bench | BBH |
| Full | 0.138 | 0.037* | < 0.001*** | 0.001** | < 0.001*** | < 0.001*** |
| Random | 0.086 | 0.354 | 0.169 | < 0.001*** | 0.878 | 0.085 |
| LIMA | 0.005** | 0.250 | 0.013* | < 0.001*** | 0.900 | < 0.001*** |
| Alpagasus | 0.079 | 0.908 | 0.099 | < 0.001*** | 0.425 | < 0.001*** |
| CaR | 0.008** | 0.354 | 0.005** | < 0.001*** | < 0.001*** | < 0.001*** |

Table 21: Statistical significance test: MP+SC at 2% (1K) vs baselines. P-values from two-sided t-test (treatment ≠ baseline). * p<0.05, ** p<0.01, *** p<0.001.

| Baseline | Mistral-7B | | Qwen3-8B | | Qwen-4B | |
| --- | --- | --- | --- | --- | --- | --- |
| | MT-Bench | BBH | MT-Bench | BBH | MT-Bench | BBH |
| Full | 0.629 | 0.417 | 0.008** | 0.001** | 0.169 | 0.120 |
| Random | 0.771 | 0.728 | 0.026* | < 0.001*** | 0.072 | < 0.001*** |
| LIMA | 0.362 | 0.016* | 0.265 | < 0.001*** | 0.034* | 0.154 |
| Alpagasus | 0.803 | 0.164 | 0.067 | < 0.001*** | 0.244 | < 0.001*** |
| CaR | 0.455 | 0.728 | 0.411 | < 0.001*** | 0.040* | 0.432 |

Table 22: Statistical significance test: MP+SC at 1% (512) vs baselines. P-values from two-sided t-test (treatment ≠ baseline). * p<0.05, ** p<0.01, *** p<0.001.

| Baseline | Mistral-7B | | Qwen3-8B | | Qwen-4B | |
| --- | --- | --- | --- | --- | --- | --- |
| | MT-Bench | BBH | MT-Bench | BBH | MT-Bench | BBH |
| Full | 0.328 | 0.246 | < 0.001*** | 0.790 | 0.011* | 0.364 |
| Random | 0.227 | 1.000 | 0.975 | < 0.001*** | 0.519 | < 0.001*** |
| LIMA | 0.018* | 0.039* | 0.244 | 0.350 | 0.347 | 0.436 |
| Alpagasus | 0.212 | 0.297 | 0.719 | < 0.001*** | 1.000 | < 0.001*** |
| CaR | 0.028* | 1.000 | 0.142 | 0.689 | 0.001** | 0.896 |

### J.3.2 GSM8K SIGNIFICANCE TESTS

Table 23: Statistical significance test for GSM8K: MP+MA vs baselines at each data percentage. P-values from two-sided t-test (treatment ≠ baseline). * p<0.05, ** p<0.01, *** p<0.001.

| Baseline | Qwen3-8B | | | | Mistral-7B | | | |
| --- | --- | --- | --- | --- | --- | --- | --- | --- |
| | 2.5% | 5% | 10% | 20% | 2.5% | 5% | 10% | 20% |
| Full | 0.801 | 0.548 | 0.445 | 0.387 | < 0.001*** | < 0.001*** | < 0.001*** | 0.011* |
| Random | 0.037* | 0.586 | 0.019* | 0.285 | 0.006** | 0.272 | 0.910 | 0.712 |
| DISF | < 0.001*** | 0.488 | 0.039* | 0.087 | 0.105 | 0.034* | 0.023* | 0.010** |
| DSIR | 0.582 | 0.767 | 0.445 | 1.000 | 0.002** | 0.071 | 0.085 | 0.006** |

Table 24: Statistical significance test for GSM8K: MP+SC vs baselines at each data percentage. P-values from two-sided t-test (treatment $\neq$ baseline). * p<0.05, ** p<0.01, *** p<0.001.

| Baseline | Qwen3-8B | | | | Mistral-7B | | | |
|---|---|---|---|---|---|---|---|---|
| | 2.5% | 5% | 10% | 20% | 2.5% | 5% | 10% | 20% |
| Full | 0.295 | 0.252 | 0.837 | 0.841 | < 0.001*** | < 0.001*** | 0.003** | 0.175 |
| Random | 0.198 | 1.000 | 0.169 | 0.684 | 0.551 | 0.905 | 0.640 | 0.116 |
| DISF | 0.001** | 0.882 | 0.277 | 0.294 | 0.603 | 0.366 | 0.009** | < 0.001*** |
| DSIR | 0.806 | 0.401 | 0.837 | 0.505 | 0.335 | 0.553 | 0.038* | < 0.001*** |

