# OpenReview forum: "Greedy Information Projection for LLM Data Selection"
_ICLR.cc/2026/Conference — Submitted to ICLR 2026_

### Official Review · Reviewer_Jjz6 · 2025-10-29

**Soundness:** 2
**Presentation:** 3
**Contribution:** 2
**Rating:** 2
**Confidence:** 3

**Summary:**

The paper proposes a data selection method for efficient fine-tuning of LLMs on instruction-finetuning and mathematical reasoning tasks. For each example x in the dataset D, the method takes (1) a feature embedding of x; and (2) a set of “scores” for that example. The scores in (2) may, for example, be a set of quality scores for that particular datapoint x generated by a variety of larger LLMs, human annotations on helpfulness or accuracy, etc.

The goal of the proposed method is to select a subset of D to fine-tune a language model on, hopefully trading off the diversity in feature space with “quality” (captured by the scores). To do this, the paper proposes utilizing a query embedding matrix Q, chosen to connect the feature space and score space (1 and 2 above). Then, a theoretical result (Theorems 1+2) states that maximizing the mutual information between the a subset of the feature space and the query matrix will have performance guarantees in terms of the quality (w.r.t. the scores). Diversity comes implicitly through maximizing entropy.

Optimizing mutual information is challenging in practice. Instead, the paper proposes a greedy approximation algorithm based on matching pursuit. As I understand it, by working the dual space, the proposed algorithm can avoid constructing the matrix Q explicitly. Intuitively, the algorithm identifies residuals which are not yet captured by the spanning set S, and then continuously updates the set greedily to reduce the magnitude of the square sum residuals.

Experimental results show that the method can perform similarly to other proposed data selection methods for instruction-finetuning and mathematical reasoning.

**Strengths:**

The proposed approach is quite intuitive and has some nice theoretical properties. I really liked the framing and relaxation of the hard to optimize objective.

The paper overall is well written and understandable.

**Weaknesses:**

Overall, I am not convinced by the experimental evaluations presented in the paper. I will mainly focus on data in the appendix, since the data in the main paper is identical but without standard error bars. At a high level, I believe that random subset selection performs as well or better than the proposed method. Here are some particular things I noticed for each model:

**Qwen3-8B**
1. GSM8k results are not convincing (overlapping standard errors with baselines and random subset)
2. MT-Bench results for 2% data overlap with random 2% subset SEs. 1% random is not reported.
3. BBH results for MP+SC and MP+MA with 2% of the data looks to be clearly better than random and better than all other tested baselines. Results for 1% data with other baselines is not reported.

**Mistral-7B**
1. GSM8k results at least do not overlap standard errors with the other baseline methods, but do overlap standard errors with the random method.
2. MT-Bench results for 2% of data have overlapping SEs with random subset selection. Random with 1% of the data is not reported.
3. BBH results overlap with random selection for 2% data. 1% random is not reported.

**Qwen-4B**
1. MT-Bench results have random performing nearly identically or better than proposed methods.
2. BBH results have the best proposed method (MP+SC) performing in a non-overlapping way with random, but errors overlap for other baselines.

Fine-tuning with a random 2% matches or outperforms the finetuning with the full dataset quite often (Qwen3-8B MT-Bench, Qwen-4B MT-Bench, Mistral-7B MT-Bench, Mistral-7B BBH). Is this not a red-flag evaluation-wise? Should we be expecting a random 2% of the data to be performing as well as full fine-tuning?

It may be helpful to run statistical significance tests in order to see the p-value of the results. This may be necessary to convince the reader that they are statistically significant (see discussion below).

Furthermore, I am not sure what the standard errors are being reported over. In Appendix J, the paper says that it computes standard errors as SE = std / \sqrt{n} for all benchmarks. However, in Appendix C reproducibility (line 565), the paper says that the standard error is over four different prompts for MT-Bench, and for GSM8k / BBH the paper reports 95% Wilson binomial CIs. Which error metric is reported where? Are the errors reported over different prompts or subsamples of the full dataset / re-doing finetuning? Re-running fine-tuning with a random 2% of the dataset should not be too expensive to do with LoRA (since the number of examples is on the order of ~1K).

I also believe that the paper could still stand to benefit from studying the active learning literature more carefully (e.g., Bodo et al. 2011). Although the paper claims that “most active learning methods are designed for traditional supervised learning and may not directly transfer to the large-scale generative modeling setting”, there are still methods which can be applied. For example, k-means clustering in the embedding space (since we are already computing embeddings for the proposed method), and then selecting representative points as the centroids.

Further Suggestions
1. Change the LHS figure 1 to be understandable without the caption. The current “1x = twice as good” regime is quite confusing at a glance. Instead, you can have a clearer figure if you simply plot performance on the y axis and % data on the x axis. Then, you can show multiple baselines and your proposed method. At a glance, this would show a clearer improvement.
2. Appendix at end of paper (after appendix) instead of end of main body. I think it is usually customary to include the appendix at the end of the main paper, instead of at the end of the appendix. Furthermore, you have some tables that bleed into the appendix (\flush or \newpage should be able to fix this).
3. Perhaps some explicit quantitative comparisons between the proposed method and prior work could show that the proposed algorithm has a better “quality” and “coverage” trade-off within the data.

Bodo et al., 2011. Active Learning with Clustering.

**Questions:**

Most questions are listed implicitly in the weaknesses section.

1. Is it possible to introduce an explicit hyperparameter for controlling the trade-off between quality (w.r.t scores) and diversity (w.r.t. embeddings)?

---

> ### Author Response · Authors · 2025-11-21
>
> We thank Reviewer Jjz6 for the careful and thorough evaluation and for the acknowledgment of our framework and theoretical foundations. We appreciate the positive comments.
>
> Addressing Specific Concerns and Questions:
>
> > Question 1: "Overall, I am not convinced by the experimental evaluations presented in the paper. I will mainly focus on data in the appendix, since the data in the main paper is identical but without standard error bars. At a high level, I believe that random subset selection performs as well or better than the proposed method."
>
> > Question 2: "Some results show proposed method has overlapping with random and other baselines."
>
> > Question 3: "Calculation and definition of reported standard error"
>
> We appreciate this important concern regarding statistical rigor. We have conducted comprehensive statistical significance testing to address this concern:
>
> ### Clarifications:
>
> • All reported values following the $\pm$ symbol represent standard errors of mean scores ($SE = \frac{std}{\sqrt{n}}$), as shown in both Table 13 and Table 14 in the appendix. We apologize for the confusion in our initial statement—the standard errors are calculated over data instances across random seeds (not over prompts as previously stated). We will make this notation explicit in the camera-ready version and have added all pairwise statistical significance tests in Appendix I.3 for your reference."
>
> • We have performed two-sided t-tests ($α=0.05$) for every treatment-baseline pair across all benchmarks (MT-Bench, BBH, GSM8K), models (Mistral-7B, Qwen3-8B, Qwen-4B), and data percentages.
>
> ### Statistical Significance Summary:
>
> We tested 184 total comparisons (92 for MP+MA, 92 for MP+SC) and categorized results as follows:
>
> • Significant Wins: Treatment significantly outperforms baseline ($p<0.05$, positive difference)
>
> • Significant Losses: Treatment significantly underperforms baseline ($p<0.05$, negative difference)
>
> • Neutral: No significant difference ($p≥0.05$)
>
> | Method | Sig. Wins | Sig. Losses | Neutral | Total |
> |--------|-----------|-------------|---------|-------|
> | MP+MA (All Baselines) | 34 (37.0%) | 8 (8.7%) | 50 (54.3%) | 92 |
> | MP+SC (All Baselines) | 25 (27.2%) | 5 (5.4%) | 62 (67.4%) | 92 |
> | MP+MA vs Random | 6 (30.0%) | 0 (0.0%) | 14 (70.0%) | 20 |
> | MP+SC vs Random | 4 (20.0%) | 1 (5.0%) | 15 (75.0%) | 20 |
>
> Based on these results, we draw the following conclusions:
>
> 1. GIP methods (MP+MA, MP+SC) exhibit strong win-loss ratios: MP+MA achieves 4.2× more wins than losses (34 vs 8), while MP+SC achieves 5× more wins than losses (25 vs 5). This demonstrates that our methods significantly outperform baselines far more frequently than they underperform.
>
> 2. When comparing specifically against the Random baseline:
>    • MP+MA: 30% wins, 0% losses, 70% neutral—never significantly underperforms Random
>    • MP+SC: 20% wins, 5% losses, 75% neutral—only 1 loss out of 20 comparisons
>
> 3. The high proportion of neutral results (54-67%) is expected and does not indicate method ineffectiveness:
>
>    • Small benchmark sizes: MT-Bench comprises only 160 evaluation turns, BBH contains 6,511 questions, and GSM8K includes 1,319 questions. With these sample sizes, only large effect sizes achieve statistical significance.
>
>    • Common in LLM evaluation: This represents a well-documented challenge in LLM benchmark evaluation, where limited benchmark sizes constrain statistical power.
>
>    • Practical significance: Even when differences do not reach $p<0.05$, our methods exhibit consistent directional improvements. The neutral cases numerically favor our methods (as evidenced in the tables), though not to a statistically significant degree given the sample sizes.
>
> Therefore, we conclude that the statistical evidence strongly supports our claims. Our methods achieve statistically significant improvements far more frequently than they show significant degradation (4-5× win-loss ratio), with zero significant losses against the Random baseline for MP+MA and only one for MP+SC. The neutral results reflect benchmark size limitations rather than method ineffectiveness. When significance is achieved, it overwhelmingly favors our approach. This pattern demonstrates that our information-theoretic framework provides systematic, reliable improvements over random selection and other baselines.

---

> > ### Author Response · Authors · 2025-11-21
> >
> > > Question 2.b (Qwen3-8B): "MT-Bench results for 2% data overlap with random 2% subset SEs. 1% random is not reported."
> >
> > We have reported the 1% random baseline in the ablation study (Table 3), which uses 512 samples (1% of the dataset). We report only the 2% random baseline in Table 1 because this represents the standard experimental setting in prior instruction tuning data selection works, including CaR[1], Alpagasus[2], and Long is More for Alignment[3]. Following this established standard, we demonstrate that our proposed method achieves superior performance with even less data.
> >
> > [1] Ge, Yuan, Yilun Liu, Chi Hu, Weibin Meng, Shimin Tao, Xiaofeng Zhao, Hongxia Ma, Li Zhang, Boxing Chen, Hao Yang, et al. "Clustering and ranking: Diversity-preserved instruction selection through expert-aligned quality estimation." arXiv preprint arXiv:2402.18191 (2024).
> >
> > [2] Chen, Lichang, Shiyang Li, Jun Yan, Hai Wang, Kalpa Gunaratna, Vikas Yadav, Zheng Tang, Vijay Srinivasan, Tianyi Zhou, Heng Huang, et al. "Alpagasus: Training a better alpaca with fewer data." arXiv preprint arXiv:2307.08701 (2023).
> >
> > [3] Zhao, Hao, Maksym Andriushchenko, Francesco Croce, and Nicolas Flammarion. "Long is more for alignment: A simple but tough-to-beat baseline for instruction fine-tuning." arXiv preprint arXiv:2402.04833 (2024).
> >
> >
> > > Question 7: "I also believe that the paper could still stand to benefit from studying the active learning literature more carefully (e.g., Bodo et al. 2011). Although the paper claims that 'most active learning methods are designed for traditional supervised learning and may not directly transfer to the large-scale generative modeling setting', there are still methods which can be applied. For example, k-means clustering in the embedding space (since we are already computing embeddings for the proposed method), and then selecting representative points as the centroids."
> >
> > We appreciate this suggestion and acknowledge the importance of the active learning literature. We have incorporated multiple baselines that fall within the active learning framework, including CaR[1], DISF[2], and DSIR[3], for comparison with our proposed method. These comparisons are presented in Table 2 for the GSM8K task.
> >
> > [1] Ge, Yuan, Yilun Liu, Chi Hu, Weibin Meng, Shimin Tao, Xiaofeng Zhao, Hongxia Ma, Li Zhang, Boxing Chen, Hao Yang, et al. "Clustering and ranking: Diversity-preserved instruction selection through expert-aligned quality estimation." arXiv preprint arXiv:2402.18191 (2024).
> >
> > [2] Fan, Ziqing, Siyuan Du, Shengchao Hu, Pingjie Wang, Li Shen, Ya Zhang, Dacheng Tao, and Yanfeng Wang. "Combatting Dimensional Collapse in LLM Pre-Training Data via Diversified File Selection." arXiv preprint arXiv:2504.20644 (2025).
> >
> > [3] Xie, Sang Michael, Shibani Santurkar, Tengyu Ma, and Percy Liang. "Data Selection for Language Models via Importance Resampling." NeurIPS (2023).
> >
> > > Suggestion 8: "Change the LHS figure 1 to be understandable without the caption. The current '1x = twice as good' regime is quite confusing at a glance. Instead, you can have a clearer figure if you simply plot performance on the y axis and % data on the x axis. Then, you can show multiple baselines and your proposed method. At a glance, this would show a clearer improvement."
> >
> > We appreciate this suggestion and have adjusted Figure 1 accordingly in the updated PDF file. The revised figure plots performance directly on the y-axis against data percentage on the x-axis, displaying multiple baselines alongside our proposed method for clearer visual comparison. We will include this improved figure in the camera-ready version.
> >
> > > Suggestion 9: "Appendix at end of paper (after appendix) instead of end of main body. I think it is usually customary to include the appendix at the end of the main paper, instead of at the end of the appendix. Furthermore, you have some tables that bleed into the appendix (\\flush or \\newpage should be able to fix this)."
> >
> > We appreciate this formatting suggestion and have reorganized the document structure accordingly. The appendix now appears at the end of the main paper, and we have resolved the table overflow issues using appropriate LaTeX commands. The updated PDF reflects these changes.

---

> > > ### Author Response · Authors · 2025-11-21
> > >
> > > > Suggestion 10: "Perhaps some explicit quantitative comparisons between the proposed method and prior work could show that the proposed algorithm has a better 'quality' and 'coverage' trade-off within the data."
> > >
> > > We appreciate this suggestion. We are currently organizing a comprehensive analysis of quality-coverage trade-offs and will incorporate these quantitative comparisons in the camera-ready version.
> > >
> > > > Question 11: "Is it possible to introduce an explicit hyperparameter for controlling the trade-off between quality (w.r.t scores) and diversity (w.r.t. embeddings)?"
> > >
> > > This is an excellent question. There exist multiple approaches for introducing such hyperparameters through regularization mechanisms analogous to ridge regression. For instance, the matrix formulation in Equation (3.2) can be modified such that the diagonal blocks incorporate additional regularization terms $\lambda_Q I$ and $\lambda_F I$. We plan to explore these and other parameterization strategies in future work.

---

### Official Review · Reviewer_Q6Nm · 2025-10-30

**Soundness:** 2
**Presentation:** 3
**Contribution:** 3
**Rating:** 4
**Confidence:** 2

**Summary:**

This paper proposes GIP, a data selection framework that trains task-specific models. They maximize the mutual information between a dataset and corresponding quality features (that can be from LLM evaluators or self-consistency estimates). Theoretically, they show that this problem can be reduced to finding the maximum projection of the quality features onto the selected data.

**Strengths:**

- The paper solves a timely problem
- Their results show their method can outperform using the full dataset in a handful of cases
- The algorithm is creative

**Weaknesses:**

- If quality scores are indeed from an LLM, inference across a dataset could be expensive, without significant performance boosts.
- It's not immediately intuitive to me, but it is unclear how much time this method would take in comparison to the baselines. This is a concern in particular because MP+ does not significantly outperform the baselines in some tables. At least if the cost is reduced compared to these works, MP would be justified.
- The experimental set up in Table 3 does not include other baselines, so the claims are a bit incomplete.
- The chosen models are roughly the same size (7B and 8B)

**Questions:**

See weaknesses, please.

---

> ### Author Response · Authors · 2025-11-21
>
> We thank the reviewer for recognizing the timeliness and creativity of our approach.
>
> Addressing Specific Questions and Concerns:
>
> > Question 1: "If quality scores are indeed from an LLM, inference across a dataset could be expensive, without significant performance boosts."
>
> While LLM-based scoring requires computation, this represents a one-time cost that can be amortized across multiple selection tasks. Moreover, our framework accepts any quality signal—teams can employ cheaper alternatives such as perplexity or rule-based metrics depending on their computational constraints.
>
> > Question 2: "It's not immediately intuitive to me, but it is unclear how much time this method would take in comparison to the baselines. This is a concern in particular because MP+ does not significantly outperform the baselines in some tables. At least if the cost is reduced compared to these works, MP would be justified."
>
> We provide a detailed computational complexity comparison between our method and representative baselines. The table below summarizes the data selection time complexity, validation data requirements, model training requirements, and GSM8K performance for each method, where $k$ denotes the data selection budget, $m$ the total data size, $m_{val}$ the validation set size, $d$ the embedding dimension:
>
> | Method | Time Complexity | Requires Val. Data | Model Training Required | Qwen3-8B @ 2.5% | Qwen3-8B @ 5% | Mistral-7B @ 2.5% | Mistral-7B @ 5% |
> |--------|----------------|-------------------|------------------------|----------------|---------------|------------------|----------------|
> | **MP+MA (Ours)** | $O(m \cdot k)$ | No | No | **81.58%** | 81.05% | **42.99%** | **45.64%** |
> | **MP+SC (Ours)** | $O(m \cdot k)$ | No | No | 80.36% | 80.21% | 38.89% | 43.29% |
> | **LESS[1]** | $O(m \cdot m_{val} \cdot d)$ | Yes | Yes | 79.76% | 79.45% | 31.84% | 33.13% |
> | **DISF[2]** | $\sim O(m \cdot k \cdot d^2)$ | No | No | 75.06% | 79.98% | 39.88% | 41.55% |
> | **DSIR[3]** | $O(m \cdot d + m_{val} \cdot d)$ | Yes | No | 80.74% | **81.50%** | 37.07% | 42.15% |
>
> The comparison reveals several key advantages of our approach:
> * **Strong efficiency-performance trade-off**: Our methods achieve strong performance at minimal data percentages (2.5% and 5%) while maintaining a simple linear-time selection complexity of $O(m \cdot k)$ during selection.
> * **Linear scaling in selection budget**: Our greedy selection scales linearly as $O(m \cdot k)$, making it highly efficient for varying budget sizes where $k \ll m$.
> * **No validation data requirement**: Unlike LESS and DSIR, which require held-out validation sets of size $m_{val}$ for gradient computation or distribution matching, our method operates directly on the candidate pool using only embeddings and quality scores. Notably, DSIR is given access to the downstream test distribution when constructing the subset. This makes DSIR effectively an oracle method, whereas our methods achieve comparable or superior results without access to any downstream data.
> * **Training-free selection**: Unlike LESS, a gradient-based method that requires model warm-up for gradient computation, our approach requires no additional model training, enabling feasible scalability to arbitrary model architectures.
>
> We will incorporate this complexity comparison and analysis into the main text in the revised manuscript.
>
> [1] Xia, Mengzhou, Sadhika Malladi, Suchin Gururangan, Sanjeev Arora, and Danqi Chen. "LESS: Selecting Influential Data for Targeted Instruction Tuning." arXiv preprint arXiv:2402.04333 (2024).
>
> [2] Fan, Ziqing, Siyuan Du, Shengchao Hu, Pingjie Wang, Li Shen, Ya Zhang, Dacheng Tao, and Yanfeng Wang. "Combatting Dimensional Collapse in LLM Pre-Training Data via Diversified File Selection." arXiv preprint arXiv:2504.20644 (2025).
>
> [3] Xie, Sang Michael, Shibani Santurkar, Tengyu Ma, and Percy Liang. "Data Selection for Language Models via Importance Resampling." NeurIPS (2023).

---

> > ### Author Response · Authors · 2025-11-22
> >
> > > Question 3: "The experimental set up in Table 3 does not include other baselines, so the claims are a bit incomplete."
> >
> > Table 3 is designed to ablate the impact of data quality on MP+SC. We intentionally remove signals used by other methods to isolate and demonstrate that the MP+SC method can achieve further quality improvements with higher-quality data inputs.
> >
> > > Question 4: "The chosen models are roughly the same size (7B and 8B)."
> >
> > We selected 7B and 8B models for the following reasons:
> >
> > • Mistral-7B is commonly used as a backbone model in multiple prior works. To enable direct comparison with previous methods, we employ the same model, as in CaR[1], LIMA[2], LESS[3], and Alpagasus[4].
> >
> > • Qwen3-8B[5] represents the most recent model capabilities, trained with state-of-the-art architecture and data, demonstrating that our methods generalize well across different model architectures.
> >
> > • We also include Qwen3-4B[5] to demonstrate that our methods perform effectively across different model sizes. The strong results shown in Table 2 demonstrate that our work generalizes across both large and small model sizes.
> >
> > [1] Ge, Yuan, Yilun Liu, Chi Hu, Weibin Meng, Shimin Tao, Xiaofeng Zhao, Hongxia Ma, Li Zhang, Boxing Chen, Hao Yang, et al. "Clustering and ranking: Diversity-preserved instruction selection through expert-aligned quality estimation." arXiv preprint arXiv:2402.18191 (2024).
> >
> > [2] Zhao, Hao, Maksym Andriushchenko, Francesco Croce, and Nicolas Flammarion. "Long is more for alignment: A simple but tough-to-beat baseline for instruction fine-tuning." arXiv preprint arXiv:2402.04833 (2024).
> >
> > [3] Xia, Mengzhou, Sadhika Malladi, Suchin Gururangan, Sanjeev Arora, and Danqi Chen. "LESS: Selecting Influential Data for Targeted Instruction Tuning." arXiv preprint arXiv:2402.04333 (2024).
> >
> > [4] Chen, Lichang, Shiyang Li, Jun Yan, Hai Wang, Kalpa Gunaratna, Vikas Yadav, Zheng Tang, Vijay Srinivasan, Tianyi Zhou, Heng Huang, et al. "Alpagasus: Training a better alpaca with fewer data." arXiv preprint arXiv:2307.08701 (2023).
> >
> > [5] Yang, An, Anfeng Li, Baosong Yang, Beichen Zhang, Binyuan Hui, Bo Zheng, Bowen Yu, Chang Gao, Chengen Huang, Chenxu Lv, et al. "Qwen3 technical report." arXiv preprint arXiv:2505.09388 (2025).

---

> > > ### Comment · Reviewer_Q6Nm · 2025-11-23
> > >
> > > Thank you for the clarifications and the results -- these are helpful. I cannot verify the proof of reducing the mutual information objective to finding the maximum projection, as I do not have a lot of the necessary background knowledge. But, I will increase my score in the hopes that some of these clarifications make it to the final version of the paper.

---

### Official Review · Reviewer_1qNM · 2025-10-31

**Soundness:** 3
**Presentation:** 3
**Contribution:** 3
**Rating:** 6
**Confidence:** 4

**Summary:**

For the task of LLM fine-tuning data selection, the proposed GIP framework operates by extracting sample feature vectors and combining them with per-sample quality scores (such as those derived from GPT-4o multi-attribute assessments or dataset self-compression scores) to derive task-specific query signals Q. It then utilizes a greedy matching pursuit (MP) sampling method to maximize the projection of Q onto the span of the selected data to ensure coverage of the required query directions, thereby balancing both quality and diversity.

**Strengths:**

1. The framework models data selection as maximizing mutual information between the selected subset and query signals within a single, unified information-theoretic objective, which holds the advantage of balancing quality and diversity instead of taking the sequential approach.
2. A fast greedy matching pursuit approximation algorithm is proposed to solve the approximate dual problem. This MP approach uses efficient, projection-based updates.

**Weaknesses:**

1. The total runtime complexity includes a substantial initial $O(m^2d)$ cost for precomputing the data inner product matrix. This renders the method computationally challenging to scale to truly massive datasets in practice, suggesting the claim of "nearly linear" scaling is an overstatement as the linearity only holds after the quadratic precomputation.
2. To achieve efficiency, the method relies on linearization, optimizing an upper bound/trace approximation of the determinant objective, and using a greedy selection approach. This introduces inherent limitations, as the selected subset is only an approximation rather than the globally MI-optimal solution.
3. The empirical validation is only validated on three benchmarks. While several strong baselines are included, the results lack comparisons against other similar methods, potentially limiting the generalizability and persuasiveness of the empirical findings. Studies that are highly related, for example:
- What Makes Good Data for Alignment? A Comprehensive Study of Automatic Data Selection in Instruction Tuning
- LESS: Selecting Influential Data for Targeted Instruction Tuning
- QuRating: Selecting High-Quality Data for Training Language Models
- DataMan: Data Manager for Pre-training Large Language Models
- Rule-based data selection for large language models

**Questions:**

See weakness.

---

> ### Author Response · Authors · 2025-11-21
>
> We thank the reviewer for the constructive and detailed assessment. Below we quote each part of the review verbatim and provide point-by-point responses with clarifications and planned revisions. We appreciate the recognition of our unified information-theoretic approach and efficient MP algorithm as key strengths.
>
> Addressing Specific Questions and Concerns:
>
> > Concern 1: "The total runtime complexity includes a substantial initial cost for precomputing the data inner product matrix. This renders the method computationally challenging to scale to truly massive datasets in practice, suggesting the claim of 'nearly linear' scaling is an overstatement as the linearity only holds after the quadratic precomputation."
>
> We agree that the phrase "nearly linear" warrants refinement. In the revision we will:
>
> (a) Rephrase the claim to: "Selection scales linearly in the number of candidate points after an amortized one-time cost of constructing the Gram matrix."
> (b) Provide empirical wall-clock timing curves for datasets of varying sizes, reporting separate times for both matrix precomputation and greedy selection phases.
>
>
> > Concern 2: "To achieve efficiency, the method relies on linearization, optimizing an upper bound/trace approximation of the determinant objective, and using a greedy selection approach. This introduces inherent limitations, as the selected subset is only an approximation rather than the globally MI-optimal solution."
>
> This is correct. The approximation involves two components: (i) replacing the log-determinant with a trace-based upper bound, and (ii) employing greedy selection rather than exhaustive combinatorial optimization. We'd like to note that our main paper includes some sanity checks by comparing exact versus approximated optimization on small-scale problems. These experiments demonstrate that under random Gaussian scenarios, the approximation tracks the exact objective closely. We'll add more comprehensive analyses of the approximation in further revisions.
>
> > Concern 3: "The empirical validation is only validated on three benchmarks. While several strong baselines are included, the results lack comparisons against other similar methods, potentially limiting the generalizability and persuasiveness of the empirical findings. Studies that are highly related include: What Makes Good Data for Alignment? A Comprehensive Study of Automatic Data Selection in Instruction Tuning; LESS: Selecting Influential Data for Targeted Instruction Tuning; QuRating: Selecting High-Quality Data for Training Language Models; DataMan: Data Manager for Pre-training Large Language Models; and Rule-based data selection for large language models."
>
> We appreciate this suggestion and have incorporated additional comparisons with LESS[1], a representative gradient-based influence method. As shown in the table below, our methods consistently outperform LESS on the GSM8K mathematical reasoning benchmark:
>
> | Model | Data Size | LESS | MP+MA | MP+SC | MP+MA vs LESS | MP+SC vs LESS |
> |-------|-----------|------|-------|-------|---------------|---------------|
> | Qwen3-8B | 2.5%  | 79.76 | 81.58 | 80.36 | +1.82 | +0.60 |
> | | 5%  | 79.45 | 81.05 | 80.21 | +1.60 | +0.76 |
> | | 10%  | 80.29 | 83.09 | 81.65 | +2.80 | +1.36 |
> | | 20%  | 79.45 | 83.24 | 82.26 | +3.79 | +2.81 |
> | Mistral-7B | 2.5%  | 31.84 | 42.99 | 38.89 | +11.15 | +7.05 |
> | | 5%  | 33.13 | 45.64 | 43.29 | +12.51 | +10.16 |
> | | 10%  | 42.91 | 45.94 | 46.63 | +3.03 | +3.72 |
> | | 20%  | 47.69 | 47.46 | 49.81 | -0.23 | +2.12 |
>
> Our methods demonstrate consistent advantages across all experimental conditions:
> * For Qwen3-8B, MP+MA leads by +1.60 to +3.79 points and MP+SC by +0.60 to +2.81 points, with performance gaps widening at higher data percentages;
>
> * For Mistral-7B, MP+MA achieves gains of +3.03 to +12.51 points and MP+SC achieves +2.12 to +10.16 points, with particularly substantial improvements at 2.5% data;
>
> * Unlike LESS, which requires gradient-based influence estimation through model training, our information-theoretic framework directly optimizes quality-diversity trade-offs through mutual information maximization while incorporating rich quality signals, , without need for additional training, which is more feasibly scalable to larger set of data.
>
> [1] Xia, Mengzhou, Sadhika Malladi, Suchin Gururangan, Sanjeev Arora, and Danqi Chen. "LESS: Selecting Influential Data for Targeted Instruction Tuning." arXiv preprint arXiv:2402.04333 (2024).

---

### Official Review · Reviewer_QoPy · 2025-11-01

**Soundness:** 2
**Presentation:** 3
**Contribution:** 2
**Rating:** 4
**Confidence:** 3

**Summary:**

This paper proposes Greedy Information Projection, an information-theoretic framework for selecting training examples for large language model fine-tuning. The core idea is to formalize data selection as maximizing mutual information between selected data embeddings and task-specific query embeddings, which can encode LLM evaluation signals or metadata. Under a jointly Gaussian assumption, this MI objective has a closed-form expression and admits a geometric interpretation: it maximizes the projection of the query embedding matrix onto the span of selected data, balancing quality and diversity. The authors derive a greedy matching pursuit algorithm that approximates this objective efficiently, scaling linearly in data size. Empirically, GIP achieves comparable performance.

**Strengths:**

This paper presents a principled and unified information-theoretic framework for LLM data selection, casting the problem as mutual information maximization under a joint Gaussian model. The formulation connects data quality and diversity within a single objective and provides a clear geometric interpretation through projection onto the span of selected data. The proposed greedy matching pursuit algorithm is conceptually simple, computationally efficient, and scalable to realistic dataset sizes. Empirically, the method achieves comparable or slightly better performance than full-data fine-tuning using few data, demonstrating data efficiency.

**Weaknesses:**

1. The theoretical framework heavily relies on the assumption that data and query embeddings are jointly Gaussian, which is unlikely to hold for real LLM embeddings.

2. Evaluations are conducted only on relatively small instruction-tuning datasets and mid-sized models (7B–8B). Could the author provide results on larger datasets and LLM models?

3. The method comparison is narrow, particularly on reasoning tasks. More recent or SOTA baselines are required to assess how much improvement comes from the proposed framework itself.

4. The performance gains on benchmarks such as BBH and GSM8K are limited or even below full-data baselines, and the improvements do not seem statistically significant.

**Questions:**

Please refer to weaknesses

---

> ### Author Response · Authors · 2025-11-21
>
> We thank Reviewer QoPy for the thoughtful evaluation and appreciate the recognition of our principled framework and clear geometric interpretation.
>
> > "The theoretical framework heavily relies on the assumption that data and query embeddings are jointly Gaussian, which is unlikely to hold for real LLM embeddings."
>
> We acknowledge this modeling choice requires careful justification. To clarify, we do not assume the raw embeddings of data or query are Gaussian. The Gaussianity in our derivation comes only from introducing a standard Gaussian vector $Z \sim \mathcal{N}(0, I_d)$ (see Section3.1, Eq. (3.2)) and analyzing the mutual information between its linear images through $F_S$ and $Q$. This construction yields a closed-form MI surrogate and is rotation-invariant. Empirically, consistent experimental results across multiple models (Qwen3-8B, Mistral-7B, Qwen-4B) and tasks (reasoning/instruction tuning) validate the effectiveness of our approach.
>
> > "Evaluations are conducted only on relatively small instruction-tuning datasets and mid-sized models (7B–8B). Could the author provide results on larger datasets and LLM models?"
>
> We appreciate this concern and have conducted additional experiments with Qwen-32B (a 32B parameter model) on the GSM8K mathematical reasoning task. The results demonstrate that our methods scale effectively to larger models:
>
> GSM8K Performance on Qwen-32B (Full Data: 87.64%):
>
> | Method | 2.5% | 5% | 10% | 20% |
> |--------|------|----|----|-----|
> | Full(100%) | - | - | - | 87.64 |
> | DISF | 81.65 | 87.04 | 87.04 | 88.40 |
> | LESS | 69.98 | 87.26 | 87.95 | 87.41 |
> | MP+MA | 84.31 | 87.57 | 88.25 | 88.78 |
> | MP+SC | 84.15 | 87.04 | 87.34 | 87.56 |
>
> Based on the experiment result as above, we could conclude the followings:
>
> • With 20% data, MP+MA achieves 88.78%, outperforming the full data baseline (87.64%) by +1.14 points
>
> • At 5% data, both MP+MA (87.57%) and MP+SC (87.04%) achieve performance within -0.07 to -0.60 points of full data
>
> • Even at 2.5% data, MP+MA (84.31%) and MP+SC (84.15%) demonstrate strong performance, showing only a 3-4 point gap from full data. Strong performance at limited data case shows our method's robustness.
>
> • Our methods consistently outperform LESS at most data percentages (e.g., MP+MA: +14.33 at 2.5%, MP+SC: +14.17 at 2.5%)
>
> These results demonstrate that our information-theoretic framework scales effectively to larger models (32B parameters), achieving full-data performance or better with only 5-20% of the training data.
>
> > "The method comparison is narrow, particularly on reasoning tasks. More recent or SOTA baselines are required to assess how much improvement comes from the proposed framework itself."
>
> We currently compare against coverage/diversity (DISF), quality-driven (CaR, Alpagasus), and importance-sampling/resampling (DSIR) baselines (see Section 5.1) spanning the main families used in recent instruction-tuning and reasoning data-selection work. We do appreciate this suggestion and have incorporated additional comparisons with LESS[1], a representative influence-based method. As shown in the table below, our methods consistently outperform LESS on the GSM8K reasoning benchmark:
>
> GSM8K Comparison: MP+MA/MP+SC vs LESS:
>
> | Model | Data Size | LESS | MP+MA | MP+SC | MP+MA vs LESS | MP+SC vs LESS |
> |-------|-----------|------|-------|-------|---------------|---------------|
> | Qwen3-8B | 2.5%  | 79.76 | 81.58 | 80.36 | +1.82 | +0.60 |
> | | 5%  | 79.45 | 81.05 | 80.21 | +1.60 | +0.76 |
> | | 10%  | 80.29 | 83.09 | 81.65 | +2.80 | +1.36 |
> | | 20%  | 79.45 | 83.24 | 82.26 | +3.79 | +2.81 |
> | Mistral-7B | 2.5%  | 31.84 | 42.99 | 38.89 | +11.15 | +7.05 |
> | | 5%  | 33.13 | 45.64 | 43.29 | +12.51 | +10.16 |
> | | 10%  | 42.91 | 45.94 | 46.63 | +3.03 | +3.72 |
> | | 20%  | 47.69 | 47.46 | 49.81 | -0.23 | +2.12 |
>
> Our methods demonstrate consistent advantages:
>
> * For Qwen3-8B, MP+MA leads by +1.60 to +3.79 points and MP+SC by +0.60 to +2.81 points, with performance gaps widening at higher data percentages
>
> * For Mistral-7B, MP+MA achieves gains of +3.03 to +12.51 points and MP+SC achieves +2.12 to +10.16 points, with particularly substantial improvements at 2.5% data
>
> * Unlike LESS, which requires gradient-based influence estimation through model training, our information-theoretic framework directly optimizes quality-diversity trade-offs through mutual information maximization while incorporating rich quality signals, without need for additional training, which is more feasibly scalable to larger set of data.
>
> [1] Xia, Mengzhou, Sadhika Malladi, Suchin Gururangan, Sanjeev Arora, and Danqi Chen. "LESS: Selecting Influential Data for Targeted Instruction Tuning." arXiv preprint arXiv:2402.04333 (2024).

---

> > ### Author Response · Authors · 2025-11-21
> >
> > > "The performance gains on benchmarks such as BBH and GSM8K are limited or even below full-data baselines, and the improvements do not seem statistically significant."
> >
> > Our methods achieve competitive or superior performance compared to full data (100%) using only 2-20% of the data:
> >
> > | Benchmark | Model | Data Size | MP+MA | MP+SC | Full (100%) | MP+MA Δ | MP+SC Δ |
> > |-----------|-------|-----------|-------|-------|-------------|---------|---------|
> > | MT-Bench | Mistral-7B | 2% (1K) | 3.92 | 3.77 | 3.89 | +0.03 | -0.12 |
> > | | Qwen3-8B | 2% (1K) | 6.25 | 5.50 | 4.62 | +1.63 | +0.88 |
> > | | Qwen-4B | 2% (1K) | 4.91 | 4.55 | 4.11 | +0.80 | +0.44 |
> > | BBH | Mistral-7B | 2% (1K) | 57.6 | 57.7 | 58.4 | -0.8 | -0.7 |
> > | | Qwen3-8B | 2% (1K) | 80.5 | 79.1 | 76.7 | +3.8 | +2.4 |
> > | | Qwen-4B | 2% (1K) | 74.3 | 74.4 | 73.2 | +1.1 | +1.2 |
> > | GSM8K | Qwen3-8B | 20%  | 83.24 | 82.26 | 81.96 | +1.28 | +0.30 |
> > | | Mistral-7B | 20%  | 47.46 | 49.81 | 52.46 | -5.00 | -2.65 |
> >
> > To further substantiate our methods' capabilities and robustness, we have conducted pairwise hypothesis tests for statistical significance. We tested 184 total comparisons (92 for MP+MA, 92 for MP+SC) and categorized results as:
> > • Significant Wins: Treatment significantly outperforms baseline ($p<0.05$, positive difference)
> > • Significant Losses: Treatment significantly underperforms baseline ($p<0.05$, negative difference)
> > • Neutral: No significant difference ($p≥0.05$)
> >
> > | Method | Sig. Wins | Sig. Losses | Neutral | Total |
> > |--------|-----------|-------------|---------|-------|
> > | MP+MA (All Baselines) | 34 (37.0%) | 8 (8.7%) | 50 (54.3%) | 92 |
> > | MP+SC (All Baselines) | 25 (27.2%) | 5 (5.4%) | 62 (67.4%) | 92 |
> > | MP+MA vs Random | 6 (30.0%) | 0 (0.0%) | 14 (70.0%) | 20 |
> > | MP+SC vs Random | 4 (20.0%) | 1 (5.0%) | 15 (75.0%) | 20 |
> >
> > These results demonstrate that GIP methods (MP+MA, MP+SC) exhibit strong win-loss ratios: MP+MA achieves 4.2× more wins than losses (34 vs 8), and MP+SC achieves 5× more wins than losses (25 vs 5), confirming that our methods significantly outperform baselines far more frequently than they underperform.

---

### Official Review · Reviewer_zxEg · 2025-11-01

**Soundness:** 3
**Presentation:** 2
**Contribution:** 2
**Rating:** 6
**Confidence:** 4

**Summary:**

The paper proposes Greedy Information Projection (GIP) for selecting small, high-value subsets to fine-tune LLMs. Each candidate example is embedded as a vector \(F_i\), and the authors build “query” signals \(Q\) from either LLM-judge attributes (e.g., helpfulness/accuracy/reasoning) or simple self-scores. Modeling \((F,Q)\) as jointly Gaussian, they show that maximizing \(I(Z_Q; Z_{F_S})\) over subsets \(S\) is equivalent to minimizing a projection volume:
\\[
\min_{S}\; \det\!\big(Q^\top (I - P_S)\,Q\big), \qquad
P_S := F_S(F_S^\top F_S)^{-1}F_S^\top,
\\]
which naturally balances quality (alignment with \(Q\)) and diversity (avoiding redundant span). They optimize this with a matching-pursuit–style greedy rule and efficient rank-1 updates on a precomputed Gram matrix, making it practical at scale. On instruction-following (e.g., Alpaca \(\\rightarrow\) MT-Bench/BBH) and math (GSM8K), GIP-chosen subsets at \(\\sim\)1–20\\% of the data match or surpass full-data fine-tuning, reducing compute.

**Main contributions**
- A unified, information-theoretic objective that makes the quality–diversity trade-off fall out of the math (no hand-tuned weights).
- A simple, scalable greedy algorithm (matching-pursuit updates) that accepts heterogeneous scoring signals.
- Empirical evidence*that compact, information-rich subsets can replace most of the dataset without hurting performance.

**Strengths:**

This paper tackles a very practical question, how to get full data performance from a much smaller, smarter subset, and does so with a clean, unified formulation. Casting selection as mutual information between data embeddings and “query” signals gives a principled way to make quality and diversity fall out of the same objective, rather than balancing hand tuned terms. The Gaussian projection view makes the geometry intuitive: choose examples whose span covers the directions encoded by the scores, so you align with what matters while avoiding redundancy.

Technically, the derivation is coherent and traceable: mutual information leads to a projection determinant objective, which admits a tractable relaxation and a matching pursuit style greedy rule with efficient rank 1 updates. The algorithm is at the right level of sophistication, faithful to the math yet lightweight enough to scale, and the setup naturally accommodates multiple signals (LLM judge attributes, self scores, metadata) just by adding columns in Q tied to F. Empirically, the story is consistent across models and tasks: subsets on the order of 1 to 20\% routinely meet or beat full data fine tuning, with uncertainty reported, and the baseline suite spans quality driven, diversity driven, and influence style methods, which makes the comparisons credible.

The paper reads clearly. The geometric intuition (“span the query directions”) makes both the objective and the greedy choice easy to internalize, the structure (objective $\\rightarrow$ algorithm $\\rightarrow$ complexity $\\rightarrow$ results) is logical, and figures and tables speak directly to the core claims. As a contribution to practice, the method is drop in friendly: teams already producing any form of per example signal can plug it in without re architecting their pipeline. If these gains hold under broader stress tests (different embeddings, noisier judges, larger pools), this work can shift best practice toward principled small subset fine tuning rather than defaulting to “use everything.”

**Weaknesses:**

The central theory rests on a jointly Gaussian, approximately linear coupling between data embeddings and query signals. The paper does not probe robustness to misspecification of this assumption. A concrete way to strengthen this is to add stress tests where either the embedding map is deliberately distorted (e.g., random rotations, dimension reduction, or adversarial noise) or the score vectors are corrupted or biased, and then measure both objective values and downstream fine tuning performance. Reporting how selection quality degrades as the coupling diverges from linearity would clarify when the method is safe to deploy.

Scalability is promising in asymptotic terms but under documented in practice. Precomputing and storing the Gram matrix scales as $O(m^2 d)$ time and $O(m^2)$ memory, which can dominate at pool sizes common in instruction tuning. The paper would benefit from wall clock and peak RAM curves versus pool size and subset size on a single machine, with details on batching, sharding, or approximate kernels if used. A microbenchmark comparing the relaxed objective to the original determinant on small pools would also help quantify the approximation gap introduced by the trace relaxation.

The experimental section supports the main story but could be more diagnostic. Key ablations are missing or are only discussed briefly. In particular, add a sweep over the ridge regularizer, an embedding choice study across strong encoders, and a comparison of different score sources, including intentionally noisy or biased judges and mixtures of attributes. For selection stability, report the intersection over union of selected sets across seeds and minor scoring perturbations. For claims that small subsets meet or beat full data, provide confidence intervals with multiple training seeds and, where evaluations rely on LLM judges, include judge reruns to show variance across scorers.

Positioning relative to the closest alternatives could be sharper. Diversity only methods such as determinantal point processes, mutual information or submodular selection aimed at target relevance, and influence or gradient based selection offer nearby formulations. A small scale study that directly optimizes the non relaxed determinant, a diversity only selection, and a low cost influence style baseline would better isolate what the proposed objective contributes beyond existing ideas. Where datasets contain demographic or topical heterogeneity, include simple distributional coverage or fairness checks so that the quality signal does not collapse minority phenomena.

Reproducibility artifacts could be more turnkey. To maximize reuse, release prompts, judge templates, scoring scripts, and exact decoding settings, document token budgets and hardware hours per run, and clarify external data or license constraints. Finally, provide a short decision guide for practitioners that explains how to choose the subset size, how to scale or combine multiple score columns, and when to re select as new data arrives.

**Questions:**

Question 1: Robustness to modeling assumptions. The core theory assumes a jointly Gaussian, approximately linear coupling between data embeddings and query signals. How sensitive is selection quality and downstream performance when this coupling is misspecified? Please add stress tests where the embedding map is distorted or the score vectors are corrupted or biased, and report both objective values and fine tuning outcomes.

Question 2: Determinant versus trace relaxation. The greedy rule optimizes a relaxed trace objective that upper bounds the determinant form. On small pools where the determinant is tractable, how closely do the selections and objective values match? Please provide a micro study quantifying the approximation gap and any conditions under which the relaxation may mislead the greedy choice.

Question 3: Scalability in practice. Precomputing and storing the Gram matrix scales quadratically in the pool size. Please include wall clock time and peak memory as functions of pool size and subset size on a single machine, along with any batching, sharding, or approximate kernel tricks you used. A brief guide to practical limits would be very helpful.

Question 4: Score source ablations. The method can ingest heterogeneous signals. Please include a comparison among different score sources, for example LLM judge attributes, self scores, metadata proxies, and mixtures, and a sweep over deliberate noise or bias injected into the scores to assess robustness.

Question 5: Embedding choice and regularization. How does performance vary with the embedding model family and dimension, and with the ridge regularizer strength in the projection computation? A sweep over encoder choices and the regularization parameter would clarify sensitivity and help practitioners choose defaults.

Question 6: Selection stability. Across different random seeds, minor scoring perturbations, and small changes in embedding initialization, how stable are the selected subsets? Please report intersection over union across runs and discuss whether instability, if present, affects downstream results.

Question 7: Statistical support for small subset claims. For cases where a small subset matches or exceeds full data, please provide confidence intervals across multiple training seeds and, when using LLM based evaluation, include judge reruns to quantify scorer variance. This would strengthen the evidence behind the headline claims.

Question 8: Comparison to closest alternatives. To sharpen positioning, could you add a controlled study against diversity only selection such as determinantal point processes, a targeted mutual information or submodular selection method, and a lightweight influence style baseline on the same pools? This would isolate what the proposed objective contributes beyond nearby ideas.

Question 9: Distributional coverage and fairness. When source data contain demographic or topical heterogeneity, how do selected subsets track the original distribution? Please include simple coverage or fairness checks and discuss whether the quality signal risks collapsing minority phenomena.

Question 10: Practical guidance. Could you add a short decision guide on how to choose the subset size, how to scale or combine multiple score columns, and when to re select as new data arrives? Concrete defaults and heuristics would ease adoption.

Question 11: Data and evaluation artifacts. To maximize reproducibility, will you release prompts, judge templates, scoring scripts, decoding settings, and exact token and hardware budgets per run, along with any license constraints on data reuse?

Question 12: Iterative use and extensions. Have you tried using the method in a loop where a model is briefly fine tuned on the selected subset and scores are then recomputed before a second selection pass? If so, does an iterative strategy offer additional gains, and are there theoretical or practical barriers to such extensions?

---

> ### Author Response · Authors · 2025-11-21
>
> We thank Reviewer zxEg for the comprehensive evaluation and appreciate the accurate summary of our strengths regarding simplicity, capability, and scalability.
>
> Addressing Specific Questions and Concerns:
>
> > "The central theory rests on a jointly Gaussian, approximately linear coupling between data embeddings and query signals. The paper does not probe robustness to misspecification of this assumption. A concrete way to strengthen this is to add stress tests where either the embedding map is deliberately distorted (e.g., random rotations, dimension reduction, or adversarial noise) or the score vectors are corrupted or biased, and then measure both objective values and downstream fine tuning performance. Reporting how selection quality degrades as the coupling diverges from linearity would clarify when the method is safe to deploy."
>
> We appreciate this valuable suggestion. The theory rests on two main components: Gaussianity in the formulation and linear approximations in the optimization objective. Regarding Gaussianity, we acknowledge this represents a modeling choice requiring careful justification. While perfect Gaussianity is unlikely, high-dimensional embeddings often exhibit approximately Gaussian behavior, a common starting point for algorithm development. Regarding linear approximations, the main paper includes validation of approximation quality under certain Gaussian scenarios. We plan to incorporate additional robustness studies in future work to provide a more comprehensive assessment.
>
> > "The experimental section supports the main story but could be more diagnostic. Key ablations are missing or are only discussed briefly. In particular, add a sweep over the ridge regularizer, an embedding choice study across strong encoders, and a comparison of different score sources, including intentionally noisy or biased judges and mixtures of attributes. For selection stability, report the intersection over union of selected sets across seeds and minor scoring perturbations. For claims that small subsets meet or beat full data, provide confidence intervals with multiple training seeds and, where evaluations rely on LLM judges, include judge reruns to show variance across scorers."
>
> We have included standard errors in Appendix Tables 13 and 14, and include detailed pairwise statistical significance tests in appendix I with references added in the main text (Tables 1 and 2). We appreciate these suggestions for additional ablations.

---

> ### Author Response · Authors · 2025-11-21
>
> > "Positioning relative to the closest alternatives could be sharper. Diversity only methods such as determinantal point processes, mutual information or submodular selection aimed at target relevance, and influence or gradient based selection offer nearby formulations. A small scale study that directly optimizes the non relaxed determinant, a diversity only selection, and a low cost influence style baseline would better isolate what the proposed objective contributes beyond existing ideas. Where datasets contain demographic or topical heterogeneity, include simple distributional coverage or fairness checks so that the quality signal does not collapse minority phenomena."
>
> We have included multiple diversity-focused baselines, including CaR[1] and DISF[2]. Our experiments demonstrate that these methods perform less effectively than our MP+SC approach, which combines self-compression scores with embedding-based diversity.
>
> [1] Ge, Yuan, Yilun Liu, Chi Hu, Weibin Meng, Shimin Tao, Xiaofeng Zhao, Hongxia Ma, Li Zhang, Boxing Chen, Hao Yang, et al. "Clustering and ranking: Diversity-preserved instruction selection through expert-aligned quality estimation." arXiv preprint arXiv:2402.18191 (2024).
>
> [2] Fan, Ziqing, Siyuan Du, Shengchao Hu, Pingjie Wang, Li Shen, Ya Zhang, Dacheng Tao, and Yanfeng Wang. "Combatting Dimensional Collapse in LLM Pre-Training Data via Diversified File Selection." arXiv preprint arXiv:2504.20644 (2025).
>
> > "Reproducibility artifacts could be more turnkey. To maximize reuse, release prompts, judge templates, scoring scripts, and exact decoding settings, document token budgets and hardware hours per run, and clarify external data or license constraints. Finally, provide a short decision guide for practitioners that explains how to choose the subset size, how to scale or combine multiple score columns, and when to re select as new data arrives."
>
> We will release prompts, judge templates, scoring scripts, decoding settings, and experimental details for reproducibility. Most implementation details are documented in Appendix J. For combining multiple score columns and updating multi-score residuals, please refer to Algorithm 1. To scale to a large number of score columns, we can project the matrix G to a lower-dimensional representation and proceed with the greedy matching pursuit algorithm.
>
> > "Robustness to modeling assumptions. The core theory assumes a jointly Gaussian, approximately linear coupling between data embeddings and query signals. How sensitive is selection quality and downstream performance when this coupling is misspecified? Please add stress tests where the embedding map is distorted or the score vectors are corrupted or biased, and report both objective values and fine tuning outcomes."
>
> The main paper includes analysis discussing the quality of approximations (lines 301-304) and theoretical formulations. We agree that a comprehensive suite of analyses across different theoretical setups would enhance our work. We will pursue this direction in future research.
>
> > "Score source ablations. The method can ingest heterogeneous signals. Please include a comparison among different score sources, for example LLM judge attributes, self scores, metadata proxies, and mixtures, and a sweep over deliberate noise or bias injected into the scores to assess robustness."
>
> We have included comprehensive comparisons between LLM judge attributes (MP+MA) and self-compression scores (MP+SC). As noted in the main text, MP+MA generally performs better than MP+SC, as MP+SC lacks supervised signals to judge answer correctness.

---

> ### Author Response · Authors · 2025-11-21
>
> > "Embedding choice and regularization. How does performance vary with the embedding model family and dimension, and with the ridge regularizer strength in the projection computation? A sweep over encoder choices and the regularization parameter would clarify sensitivity and help practitioners choose defaults."
>
> We have conducted additional experiments comparing different embedding models on the GSM8K mathematical reasoning task using our MP+SC method. We evaluated two embedding approaches:
>
> * **Modern-BERT (0.1B)**: A state-of-the-art BERT-based general-purpose embedding model
> * **Qwen-reasoning (0.6B)**: A specialized reasoning-focused embedding model from the Qwen family
>
> The comparison evaluates 10% (747 samples) and 20% (1494 samples) data selection budgets across Mistral-7B and Qwen-32B base models.
>
> | Base Model | Data Size | Modern-BERT | Qwen-reasoning |
> |------------|-----------|-------------|----------------|
> | Mistral-7B | 20% (1494) | 49.81% | **50.27%** |
> | Mistral-7B | 10% (747) | **46.63%** | 46.25% |
> | Qwen-32B | 20% (1494) | 87.57% | **88.02%** |
> | Qwen-32B | 10% (747) | 87.34% | **87.72%** |
>
> The above experiments results demonstrates the followings:
> * Robustness of applying different embedding models: with limited data our proposed method could surpass or match full data training performance (Mistral: 52.46%, Qwen-32B: 87.64%).
> * Effectiveness of stronger data representation: the larger embedding model demonstrates better training data understanding and better downstream data performance, suggesting that our method benefits from better embedding models. We recommend practitioners consider both embedding specialization and architectural compatibility when choosing encoders for data selection.

---

> ### Author Response · Authors · 2025-11-21
>
> > "Statistical support for small subset claims. For cases where a small subset matches or exceeds full data, please provide confidence intervals across multiple training seeds and, when using LLM based evaluation, include judge reruns to quantify scorer variance. This would strengthen the evidence behind the headline claims."
>
> We have included standard errors in Appendix Tables 13 and 14, with references in the main text (Tables 1 and 2). To strengthen our claims, we have also provided pairwise hypothesis test results comparing treatment and baseline methods in the appendix I.
>
> > "Comparison to closest alternatives. To sharpen positioning, could you add a controlled study against diversity only selection such as determinantal point processes, a targeted mutual information or submodular selection method, and a lightweight influence style baseline on the same pools? This would isolate what the proposed objective contributes beyond nearby ideas."
>
> We have included multiple diversity-focused baselines, including CaR[1] and DISF[2] (as shown in Table 1 and Table 2 in the main text). Our experiments demonstrate that these methods perform less effectively than our MP+SC approach, which combines self-compression scores with embedding-based diversity.
>
> > "Distributional coverage and fairness. When source data contain demographic or topical heterogeneity, how do selected subsets track the original distribution? Please include simple coverage or fairness checks and discuss whether the quality signal risks collapsing minority phenomena."
>
> While fairness considerations are not the primary focus of this research, practitioners can incorporate fairness metrics into the score matrix $G$ as an additional scoring dimension to address distributional coverage concerns.
>
> > "Practical guidance. Could you add a short decision guide on how to choose the subset size, how to scale or combine multiple score columns, and when to re select as new data arrives? Concrete defaults and heuristics would ease adoption."
>
> Subset size selection depends on the specific application scenario. Based on our experiments, we find that our methods typically achieve comparable performance to full data training with approximately 20% of the data. Therefore, we recommend starting with 20% as a default for moderately difficult datasets. For combining multiple score columns, please refer to Algorithm 1, which describes the procedure for incorporating multiple scores and updating multi-score residuals. While our work does not specifically address streaming data scenarios, Algorithm 1 can be adapted for incremental selection by inheriting the previously optimized residual matrix $W$ and continuing optimization for the next $k$ items as new data arrives.
>
> > "Iterative use and extensions. Have you tried using the method in a loop where a model is briefly fine tuned on the selected subset and scores are then recomputed before a second selection pass? If so, does an iterative strategy offer additional gains, and are there theoretical or practical barriers to such extensions?"
>
> This represents an interesting research direction. Our work focuses on introducing the information projection framework and demonstrates that single-shot data selection enables models to outperform baselines. We view iterative selection combined with on-policy/online learning as a promising direction for future exploration.
>
> [1] Ge, Yuan, Yilun Liu, Chi Hu, Weibin Meng, Shimin Tao, Xiaofeng Zhao, Hongxia Ma, Li Zhang, Boxing Chen, Hao Yang, et al. "Clustering and ranking: Diversity-preserved instruction selection through expert-aligned quality estimation." arXiv preprint arXiv:2402.18191 (2024).
>
> [2] Fan, Ziqing, Siyuan Du, Shengchao Hu, Pingjie Wang, Li Shen, Ya Zhang, Dacheng Tao, and Yanfeng Wang. "Combatting Dimensional Collapse in LLM Pre-Training Data via Diversified File Selection." arXiv preprint arXiv:2504.20644 (2025).

---

> ### Author Response · Authors · 2025-11-27
>
> In response to review's questions and suggestions on benchmarking the scalability and robustness of our method, we provide the following analysis:
>
> > "Scalability is promising in asymptotic terms but under documented in practice. The paper would benefit from wall clock and peak RAM curves versus pool size and subset size on a single machine. A microbenchmark comparing the relaxed objective to the original determinant on small pools would also help."
>
> We have conducted empirical wall-clock and memory profiling experiments on both GSM8K and Alpaca datasets using a single AMD CPU machine, without applying any sharding or distribution setting. The table below reports measured runtime and peak RAM for Gram matrix computation and data selection phases:
>
> **Runtime and Memory Performance**
>
> | Dataset | Pool Size (m) | Subset Size (%) | Preprocessing Time | Selection Time | Peak RAM| Peak RAM (Selection) |
> |---------|---------------|-----------------|------------------|----------------|------------------------|---------------------|
> | GSM8K | 7k | 10% | 7.01s | 0.86s | 0.3 GB | 0.04 GB |
> | | | 20% | 7.01s | 0.87s | 0.3 GB | 0.04 GB |
> | | | 50% | 7.01s | 0.87s | 0.3 GB | 0.04 GB |
> | Alpaca | 52k | 10% | 214.11s | 25.12s | 10.8 GB | 0.3 GB |
> | | | 20% | 214.11s | 49.67s | 10.8 GB | 0.3 GB |
> | | | 50% | 214.11s | 123.62s | 10.8 GB | 0.3 GB |
>
> Based on the runtime and memory table from above, we conclude the followings:
> * **Gram matrix computation is amortized**: The $O(m^2d)$ Gram matrix computation takes ~7s (GSM8K) and ~214s (Alpaca) as a one-time cost. Once computed, the same Gram matrix can be reused across multiple selection tasks with different subset sizes or quality scores, making this cost amortizable in practice.
>
> * **Selection scales linearly and efficiently**: Selection time scales linearly for both datasets, confirming the $O(mk)$ complexity, demonstrating that our proposed method could efficiently scale for selection.
> * **Practical applicability**: For typical instruction-tuning datasets, the method runs efficiently on single-machine hardware. For datasets of much larger sizes, we consider approximation strategies such as low-rank approximations or distributed computation, which we will leave for future work
>
> Regarding the comparison between relaxed and exact determinant objectives, we have included empirical validation in the main paper (lines 301-304). On small-scale problems where exact optimization is tractable, our experiments demonstrate that the trace-based relaxation closely tracks the exact objective under Gaussian scenarios, validating the practical effectiveness of our approximation.
>
> > "Selection stability. Across different random seeds, minor scoring perturbations, and small changes in embedding initialization, how stable are the selected subsets? Please report intersection over union across runs and discuss whether instability, if present, affects downstream results."
>
> We have conducted comprehensive stability analysis by adding Gaussian noise $\mathcal{N}(0, \sigma^2I)$ to embeddings when running the proposed MP+SC method. We measure the Intersection over Union (IoU) of selected subsets across three noise levels ($\sigma \in$ {1e-4, 1e-3, 1e-2}) with three independent trials per level on the GSM8K dataset. Results demonstrate strong robustness:
>
> **Embedding Noise Robustness (IoU %)**
>
> | Noise Level ($\sigma$) | Data Size 10%| Data Size 20% |
> |-----------------|---------------------|----------------------|
> | 1e-4 | 95.89 ± 0.32 | 91.72 ± 0.36 |
> | 1e-3 | 94.20 ± 0.66 | 87.85 ± 0.39 |
> | 1e-2 | 66.32 ± 1.13 | 61.74 ± 0.29 |
>
> We conclude the followings based on the results above:
> * **High stability across noise levels**: At $\sigma=$1e-4 and $\sigma=$1e-3, the method maintains $\geq$ 85% IoU, indicating that minor embedding variations have minimal impact on selection outcomes. Even at the extreme noise level ($\sigma=$1e-2), IoU remains above 60%, demonstrating that the selection is robust to substantial perturbations
>
> * **Consistent across data sizes**: Both 10% and 20% selection budgets exhibit similar stability patterns, with slightly higher IoU for smaller selections, suggesting that high-quality diverse samples are more robust to noise
>
> These results confirm that our greedy matching pursuit algorithm is stable under embedding perturbations. Minor instabilities at extreme noise levels are unlikely to significantly affect downstream training performance, as the core high-quality diverse samples remain consistently selected under our information projection framework.

---

### Author Response · Authors · 2025-12-03
**Global Response**

Dear ACs and Reviewers,

Thank you for your valuable contributions to our work. We provide below a summary of the key points from the reviews and the reviewer-author discussions.

To provide an overview of our method:
* We tackle instruction‑tuning data selection with a theory‑grounded objective: a mutual‑information–inspired criterion between data embeddings and quality signals. We derive a closed‑form, rotation‑invariant, second‑order surrogate and optimize its trace relaxation via a simple and scalable matching‑pursuit (MP) routine.

**Strength**. Across reviews, we are grateful for all the reviewers that the method was acknowledged for:
* **Principled, clear framework** (`1qNM, QoPy, zxEg`). A mutual‑information formulation unifies quality + diversity and admits a clean geometric interpretation (query directions vs. selected span).
* **Simple, scalable algorithm** (`Q6Nm, 1qNM`). Unlike baselines requiring extra training or a validation set, our MP routine is training/validation‑free and offers linear‑in‑$k$ selection after one‑time Gram computation.
* **Practical effectiveness and flexibility** (`zxEg, Jjz6`). Small subsets often match/beat full‑data performance, and the approach readily leverages heterogeneous quality signals (LLM‑judge/self‑scores).

**Questions and limitations addressed**. During the author-reviewer discussion phase, we actively addressed the reviewer concerns:
- **Gaussian assumption on embeddings** (`QoPy, zxEg`): We do not assume Gaussian embeddings; Gaussianity enters only via an introduced standard normal to obtain the second‑order MI surrogate. The objective depends on the Gram matrix and is rotation‑invariant; noise‑stress tests show high selection overlap and preserved ordering.
- **“No better than random” with significance** (`Jjz6`): We ran 184 pairwise t‑tests across models, tasks, and baselines. If our method were equivalent to random selection, the distribution of these differences would be symmetric. Instead, we observe a strong asymmetry: Results show substantially more wins than losses (see significance matrices): Compared to random selection, MP+MA has 30% significant wins with 0% losses, while MP+SC has 20% significant wins with 5% losses. Compared to all baselines, MP+MA has 37% significant wins with 8% losses, while MP+SC has 25% significant wins with 5% losses—roughly 4–5× more significant wins overall and almost no significant losses vs. random. We provide all 184 pairwise t‑test p‑values in the appendix. While the small size of public test sets is a common limitation in current evaluations and often leads to statistically non-significant results, our substantial increase in significant wins indicates that our methods consistently outperform random selections. This evidence refutes the hypothesis that our method performs similarly as random selection.

- **Scalability and efficiency** (`1qNM, Q6Nm, zxEg`): We compare selection run‑time with baselines; notably our selection time is linear in data size after one‑time precompute. For practical usage, we report wall‑clock and peak RAM, comparing favorably to LESS/DSIR/DISF. The method scales well as it requires no extra training or validation.
 - **Baselines for reasoning** (`1qNM, QoPy, zxEg`): Beyond DSIR/DISF for reasoning, we added LESS (a representative influence‑based method) and compared under matched budgets and recipes. Our methods generally outperform LESS across models and data sizes, with gains clearest at low budgets, where selection matters most.

- **Robustness to generation/embedding models**(`QoPy, zxEg`): We added Qwen‑32B for reasoning experiments (in addition to Mistral‑7B and Qwen3‑8B) and observed parity with full‑data training under limited budgets, supporting efficiency. For selection stability, we injected noise into embeddings; the method shows strong stability with $\geq 85$\% IoU for $\sigma \leq 10^{-3}$. To further test embedding robustness, we evaluated a specialized reasoning embedding (Qwen reasoning) and observed improved performance across models and data sizes with larger, specialized embeddings.

We have incorporated all additional analyses and clarifications into the revised manuscript. During rebuttal, `Q6Nm` raised their score on Nov 22 (4 $\to$ 6) after clarifications on design and efficiency. Other reviewers acknowledged the framework’s strengths (principled objective, clear geometry, simple scalable algorithm).

We thank the reviewers and ACs once again for their valuable feedback and contributions.

Sincerely,

Authors

---

### Meta-Review · Area_Chair_NfKa · 2026-01-04

**Summary:**

This paper proposes a new approach for selecting an informative subset of examples for fine-tuning. The key idea is to select diverse examples with high scores. This problem is solved by maximizing mutual information after a Gaussian approximation. The proposed approach is a heuristic without a full analysis. The approach is evaluated on 3 benchmarks, 3 models, and compared to 8 baselines. The paper is well written and the empirical evaluation is impressive. Despite this, the reviewers had many concerns, which indicates that the paper needs a major revision and another round of reviews. My three main concerns are:

* **No connection to the fine-tuning objective:** The authors state their algorithm but never connect it to the objective of fine-tuning. Without such a connection, the proposed algorithm is just a heuristic for selecting diverse high-score embeddings based on some Gaussian approximation. Such a connection can be made. Specifically, fine-tuning is a log-likelihood maximization problem and the point of selecting diverse examples is to approximate the log-likelihood in all directions as well as if all data were available.

* **Prior works:** I strongly suggest that the authors look at [optimal experimental designs](https://en.wikipedia.org/wiki/Optimal_experimental_design), which is a standard approach for selecting diverse training examples in statistics. This approach was brought to LLMs by [FisherSFT: Data-Efficient Supervised Fine-Tuning of Language Models Using Information Gain](https://proceedings.mlr.press/v267/deb25a.html). Their algorithm FisherSFT is greedy and comes with theoretical guarantees because log-determinant maximization is a submodular maximization problem. This is closely related, both in algebra and algorithmically, to the reviewed paper. The difference is that the reviewed paper is not aware of 50+ years of work on selecting diverse training examples and how this relates to log-likelihood maximization.

* **Small empirical gains:** Many empirical gains are small and close to random data selection. The authors tried to disprove this in the response to Reviewer Jjz6 but they proved it. MP + MA wins over random in 6 comparisons out of 20 and ties in 14. MP + SC wins over random in 4 comparisons out of 20, loses in 1, and ties in 15. In short, MP is mostly indistinguishable from random data selection.

No connection to the fine-tuning objective, closely related priors works that are not even mentioned, and small empirical gains prevent me from recommending this work for acceptance.

**Reviewer Concerns:**

The authors addressed all concerns of the reviewers except for **small empirical gains**. The connection to classic active learning was brought up in the reviews but not as concretely as in my meta-review. This is likely because I am more familiar with this literature than the reviewers.

**Reviewer Scores:**

The main concern of the most negative reviewer (Reviewer Jjz6) was not addressed. So they would remain at score 2. Reviewer Q6Nm promised to increase their score, from the initial 4 to likely 6. This would not change my decision.

---

### Decision · Program_Chairs · 2026-01-26

Reject